# Revisiting Robustness in Graph Machine Learning

**Lukas Gosch, Daniel Sturm, Simon Geisler, Stephan Günnemann**
Department of Computer Science & Munich Data Science Institute
Technical University of Munich
{l.gosch, da.sturm, s.geisler, s.guennemann}@tum.de

## Abstract

Many works show that node-level predictions of Graph Neural Networks (GNNs) are unrobust to small, often termed adversarial, changes to the graph structure. However, because manual inspection of a graph is difficult, it is unclear if the studied perturbations always preserve a core assumption of adversarial examples: that of unchanged semantic content. To address this problem, we introduce a more principled notion of an adversarial graph, which is aware of semantic content change. Using Contextual Stochastic Block Models (CSBMs) and real-world graphs, our results uncover: $i$) for a majority of nodes the prevalent perturbation models include a large fraction of perturbed graphs violating the unchanged semantics assumption; $ii$) surprisingly, all assessed GNNs show *over-robustness* - that is robustness *beyond* the point of semantic change. We find this to be a complementary phenomenon to adversarial robustness related to the small degree of nodes and their class membership dependence on the neighbourhood structure.

## 1 Introduction

Graph Neural Networks (GNNs) are seen as state of the art for various graph learning tasks (Hu et al., 2021). However, there is strong evidence that GNNs are unrobust to changes to the underlying graph (Zügner et al., 2018; Geisler et al., 2021). This has led to the general belief that GNNs are vulnerable to adversarial examples and many works trying to increase their robustness through various defenses (Günnemann, 2022). Originating from the study of deep image classifiers (Szegedy et al., 2014), an adversarial example has been defined as a small perturbation, often measured using an $\ell_p$-norm, which does not change the semantic content (i.e. category) of an image, but results in a different prediction. These perturbations are often termed unnoticeable relating to a human observer for whom a normal and an adversarially perturbed image are nearly indistinguishable (Goodfellow et al., 2015; Papernot et al., 2016). However, compared to visual tasks, it is difficult to visually inspect (large-scale) graphs. This has led to a fundamental question:

*What constitutes a small, semantics-preserving perturbation to a graph?*

The de facto standard in the literature is to measure small changes to the graph's structure using the $\ell_0$-pseudonorm (Günnemann, 2022). Then, the associated threat models restrict the total number of inserted and deleted edges globally in the graph and/or locally per node. However, if the observation of semantic content preservation for these kind of threat models transfers to the graph domain can be questioned: Due to the majority of low-degree nodes in real-world graphs, small $\ell_0$-norm restrictions still allow to completely remove a significant number of nodes from their original neighbourhood. Only few works introduce measures beyond $\ell_0$-norm restrictions. In particular, it was proposed to additionally use different global graph properties as a proxy for unnoticability, such as the degree distribution (Zügner et al., 2018), degree assortativity (Li et al., 2021), or other homophily metrics (Chen et al., 2022) (for further related work, see Appendix G). While these are important first steps, the exact relation between preserving certain graph properties and the graph's semantic content

2022 Trustworthy and Socially Responsible Machine Learning (TSRML 2022) co-located with NeurIPS 2022.

(e.g. node categories) is unclear. For instance, one can completely rewire the graph by iteratively exchanging nodes of two randomly selected edges and preserve the global degree distribution. As a result, current literature lacks a principled understanding of semantics preservation in their employed notions of smallness as well as robustness studies using threat models only including provable semantics-preserving perturbations to a graph. We bridge this gap by being the first to directly address the problem of exactly measuring (node-level) semantic content preservation in a graph under structure perturbations. Surprisingly, using Contextual Stochastic Block Models (CSBMs), this leads us to discover a novel phenomenon: GNNs show strong robust-

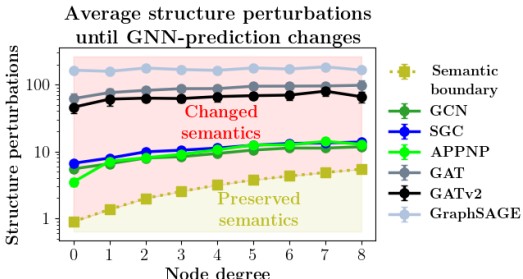

Figure 1: Data from CSBM graphs. The semantic boundary indicates when the semantic-content (i.e., the most likely class) of a node of particular degree changes on average. All GNNs show robustness *beyond* the point of semantic change.

ness *beyond* the point of semantic change (see Figure 1). This does *not* contradict the existence of adversarial examples for the same GNNs. Related to the small degree of nodes, we find that common perturbation sets include both: graphs which are truly adversarial as well as graphs with changed semantic content. Our contributions are:

1. We define a semantics-aware notion of adversarial robustness (Section 3) for node-level predictions. Using this, we introduce a novel concept: *over-robustness* - that is (unwanted) robustness against admissible perturbations with changed semantic content (i.e. changed ground-truth labels).

2. Using CSBMs, we find: $i$) common perturbations sets, next to truly adversarial examples, include a large fraction of graphs with changed semantic content (Section 4.1); $ii$) all examined GNNs show significant *over-robustness* to these graphs (Section 4.2) and we observe similar patterns on real-world datasets (Section 4.2.1). Using $\ell_0$-norm bounded adversaries on CSBM graphs, we find a considerable amount of a conventional adversarial robustness to be in fact over-robustness.

## 2 Preliminaries

Let $\mathbf{X} \in \mathbb{R}^{n \times d}$ be the node feature matrix with $n$ nodes having $d$ features, $\mathbf{A} \in \{0,1\}^{n \times n}$ the (symmetric) adjacency matrix, and $y \in \{0,1\}^n$ the node labels of which $y_L \in \{0,1\}^l$, $l \leq n$ are known. Due to the non-i.i.d data generation, a node-classifier $f$ may depend its decision on the whole known graph. Thus, we write $f(\mathbf{X}, \mathbf{A}, y_L)_v$ to denote the classification of a node $v$. We study (inductive) evasion attacks, i.e. an adversary $\mathcal{A}$, given a clean graph, can choose a perturbed graph from the perturbation set $\mathcal{B}(\mathbf{X}, \mathbf{A})$ to change the predictions of a given node-classifier $f$. We focus on direct structure attacks (Zügner et al., 2018), i.e. $\mathcal{A}$ can remove or add at most $\Delta$ edges incident to a target node $v$. We leverage *Contextual Stochastic Block Models* (CSBMs) (Deshpande et al., 2018) to generate synthetic graphs with analytically tractable distributions. It defines edge-probabilities $p$ between same-class nodes and $q$ between different-class nodes. Node-features are drawn from a Gaussian mixture model. Sampling from a CSBM can be understood as an iterative process over nodes $i \in [n]$: 1) Sample label $y_i \sim \text{Ber}(1/2)$ (Ber denoting the Bernoulli distribution). 2) Sample feature vector $\mathbf{X}_{i,:} | y_i \sim \mathcal{N}((2y_i - 1)\mu, \sigma\mathbf{I})$ with $\mu \in \mathbb{R}^d$, $\sigma \in \mathbb{R}$. 3) For all $j \in [n], j < i$ sample $\mathbf{A}_{j,i} \sim \text{Ber}(p)$ if $y_i = y_j$ and $\mathbf{A}_{j,i} \sim \text{Ber}(q)$ otherwise, and set $\mathbf{A}_{i,j} = \mathbf{A}_{j,i}$. We denote this process $(\mathbf{X}, \mathbf{A}, y) \sim \text{CSBM}_{n,p,q}^{\mu,\sigma^2}$. Inductively adding $m$ nodes can be performed by repeating the above process for $i = n+1, \ldots, m$. Fountoulakis et al. (2022) show that depending on the distance between the class means $\delta$, there is an easy regime, where a linear classifier ignoring $\mathbf{A}$ can perfectly separate the data and a hard regime, defined by $\delta = K\sigma$, with $0 < K \leq \mathcal{O}(\sqrt{\log n})$, where this is not possible.

## 3 Revisiting adversarial perturbations

Given a clean graph $(\mathbf{X}, \mathbf{A}, y_L)$, target node $v$ and perturbation set $\mathcal{B}(\mathbf{X}, \mathbf{A})$. An adversary $\mathcal{A}$ tries to choose a perturbed graph $(\tilde{\mathbf{X}}, \tilde{\mathbf{A}}) \in \mathcal{B}(\mathbf{X}, \mathbf{A})$, with the goal to change the prediction of a node classifier $f$, i.e., $f(\tilde{\mathbf{X}}, \tilde{\mathbf{A}}, y_L)_v \neq f(\mathbf{X}, \mathbf{A}, y_L)_v$. The prevalent works implicitly assume that every $(\tilde{\mathbf{X}}, \tilde{\mathbf{A}}) \in \mathcal{B}(\mathbf{X}, \mathbf{A})$ preserves the node-level semantic content of the clean graph, i.e. the original

ground-truth label of $v$. If we would have an oracle $\Omega$ telling us the semantic content, this assumption can be made explicit by writing $\Omega(\tilde{\mathbf{X}}, \tilde{\mathbf{A}}, y_L)_v = \Omega(\mathbf{X}, \mathbf{A}, y_L)_v$. In practice, we do not have access to $\Omega$, but we can try to model its behaviour by introducing a reference or base node classifier $g$. Then, the idea is to use $g$ to indicate semantic content change and thereby, define the semantic boundary (see Figure 1). Exemplary, $g$ could be derived from knowledge about the data generating process. We do so in Section 4, where we use the Bayes classifier for CSBMs as $g$ (see also Section 3.1). Note that labels themselves are often generated following a base classifier. Exemplary, this can be humans labelling a dataset. Using a reference classifier $g$ as a proxy for semantic content enables us to make a refined definition of an adversarial graph, which makes the unchanged-semantics assumption explicit:

**Definition 1.** *Let $f$ be a node classifier and $g$ a reference node classifier. Then the perturbed graph $(\tilde{\mathbf{X}}, \tilde{\mathbf{A}}) \in \mathcal{B}(\mathbf{X}, \mathbf{A})$ chosen by an adversary $\mathcal{A}$ is said to be adversarial for $f$ at node $v$ w.r.t. the reference classifier $g$ if the following conditions are satisfied:*

    *i.*    $f(\mathbf{X}, \mathbf{A}, y_L)_v = g(\mathbf{X}, \mathbf{A}, y_L)_v$      *(correct clean prediction)*

    *ii.*   $g(\tilde{\mathbf{X}}, \tilde{\mathbf{A}}, y_L)_v = g(\mathbf{X}, \mathbf{A}, y_L)_v$      *(perturbation preserves semantics)*

    *iii.*   $f(\tilde{\mathbf{X}}, \tilde{\mathbf{A}}, y_L)_v \neq g(\mathbf{X}, \mathbf{A}, y_L)_v$      *(node classifier changes prediction)*

Definition 1 says that a perturbed graph $(\tilde{\mathbf{X}}, \tilde{\mathbf{A}}) \in \mathcal{B}(\mathbf{X}, \mathbf{A})$ only then is adversarial, if $(\tilde{\mathbf{X}}, \tilde{\mathbf{A}})$ does not only change the prediction of the node classifier $f$ (iii), but also lets the original label unchanged (ii). The first constraint (i) stems from the fact that if $f$ and $g$ disagree on the clean graph at node $v$ this should represent a case of misclassification captured by standard error metrics such as accuracy.

Suggala et al. (2019) use the concept of a reference classifier in similar spirit, to define semantics-aware adversarial perturbations for i.i.d. data, with a focus on the image domain. However, what has not been considered so far, is that the reference classifier allows us to characterize the exact opposite behaviour of an adversarial example: If a classifier $f$ does not change its prediction for a perturbed graph $(\tilde{\mathbf{X}}, \tilde{\mathbf{A}})$ even though the semantic content has changed. This is undesirable, as this would mean that $f$ is robust beyond the point of semantic change. Thus, we call this behaviour *over-robustness*:

**Definition 2.** *Let $f$ be a node-classifier and $g$ a reference classifier. Then the perturbed graph $(\tilde{\mathbf{X}}, \tilde{\mathbf{A}}) \in \mathcal{B}(\mathbf{X}, \mathbf{A})$ chosen by an adversary $\mathcal{A}$ is said to be an over-robust example for $f$ at node $v$ w.r.t. the reference classifier $g$ if the following conditions are satisfied:*

    *i.*    $f(\mathbf{X}, \mathbf{A}, y_L)_v = g(\mathbf{X}, \mathbf{A}, y_L)_v$      *(correct clean prediction)*

    *ii.*   $g(\tilde{\mathbf{X}}, \tilde{\mathbf{A}}, y_L)_v \neq g(\mathbf{X}, \mathbf{A}, y_L)_v$      *(perturbation changes semantics)*

    *iii.*   $f(\tilde{\mathbf{X}}, \tilde{\mathbf{A}}, y_L)_v = g(\mathbf{X}, \mathbf{A}, y_L)_v$      *(node classifier stays unchanged)*

*If there exists such an over-robust example, we call $f$ over-robust at node $v$ w.r.t. $g$.*

Definition 2 may be of particular interest in the graph domain, where perturbation sets $\mathcal{B}(\mathbf{X}, \mathbf{A})$ often include graphs $(\tilde{\mathbf{X}}, \tilde{\mathbf{A}})$ which allow significant changes to the neighbourhood structure of $v$, but do not allow easy manual content inspections. Indeed, Section 4 shows that all assessed GNNs are over-robust for common choices of $\mathcal{B}(\mathbf{X}, \mathbf{A})$ for many test nodes $v$ in CSBM graphs. This may not be as relevant in the image domain, where small $\ell_p$-norm perturbations are visually imperceptible.

Now, we develop a deeper conceptual understanding of over- compared to adversarial robustness. In Figure 2 the decision boundary of a classifier $f$ follows the one of a base classifier $g$ except for the dotted line. The dashed region between $f$'s and $g$'s decision boundary is a region of over-robustness for the blue class and a region of adversarial examples for the red class. In practice, the perturbation sets $\mathcal{B}(\cdot)$ are bounded. As a result, it is only possible to measure the right boundary of the dashed area using adversarial examples. The concept of over-robustness allows us to additionally measure the left boundary and hence, provides us with a more complete picture of the robustness of $f$. Note that the blue data points, using conventional adversarial robustness, are judged robust in the whole of $\mathcal{B}(\cdot)$ even though their true class changed. Semantic-aware adversarial robustness (Definition 1) allows us to (correctly) cut off $\mathcal{B}(\cdot)$ at the decision boundary of $g$. We further conceptually and analytically develop other interesting cases in Appendix A.

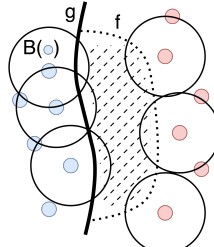

Figure 2: The decision boundary of $f$ follows base classifier $g$ except (wrongly) for the dotted line. Finite perturbation sets $\mathcal{B}(\cdot)$ intersect only from one side with the dashed area.

### 3.1 Bayes optimal classifier as reference classifier

We derive a Bayes optimal classifier for inductive node classification to use as a reference classifier $g$. This is a natural choice, because it provides us with the information, if another class is now more likely based on the true data generating distribution. We assume that the training graph is fully labeled, i.e. $y_L = y$ and focus on the most simple case of classifying an inductively sampled node. We denote the conditional distribution over graphs with an inductively added node as $\mathcal{D}(\mathbf{X}, \mathbf{A}, y)$. How well our classifier $f$ generalizes to a newly added node $v$ is captured by the expected $0/1$-loss:

$$\mathbb{E}_{(\mathbf{X}', \mathbf{A}', y') \sim \mathcal{D}(\mathbf{X}, \mathbf{A}, y)} [\ell_{0/1}(y'_v, f(\mathbf{X}', \mathbf{A}', y)_v] \tag{1}$$

To derive Bayes optimality, we have to find an optimal classifier $f^*$ for $v$, depending on $(\mathbf{X}', \mathbf{A}', y)$, minimizing (1). The following theorem shows that, similar to inductive classification for i.i.d. data, $f^*$ should choose the most likely class based on the seen data (Proof in Appendix B.1):

**Theorem 1.** *The Bayes optimal classifier, minimizing the expected 0/1-loss (1), is* $f^*(\mathbf{X}', \mathbf{A}', y)_v = \arg\max_{\hat{y} \in \{0,1\}} \mathbb{P}[y'_v = \hat{y} | \mathbf{X}', \mathbf{A}', y]$.

## 4 Results

Using Contextual Stochastic Block Models (CSBMs) we measure the extent of semantic content violations in common threat models (Section 4.1). Then, we study over-robustness in CSBMs (Section 4.2) and real-world graphs (Section 4.2.1). In CSBMs, we use the Bayes optimal classifier (Theorem 1), denoted $g$, to measure semantic change. The robustness of the Bayes classifier defines the maximal meaningful robustness achievable (see Section 3).

**Experimental setup.** We sample training graphs with $n = 1000$ nodes from several $\text{CSBM}_{n,p,q}^{\mu,\sigma^2}$ in the hard regime (Section 2). Each element of the class mean vector $\mu \in \mathbb{R}^d$ is set to $K\sigma/2\sqrt{d}$, resulting in a distance between the class means of $K\sigma$. We set $\sigma = 1$ and vary $K$ from close to *no* discriminative features $K = 0.1$ to making structure information *unnecessary* $K = 5$ (see Table 1). We choose $p = 0.63\%$, $q = 0.15\%$ resulting in the expected number of same-class and different-class edges for a given node to fit CORA (Sen et al., 2008). We set $d = \lfloor n/\ln^2(n) \rceil = 21$ (Fountoulakis et al., 2022). We use an $80\%/20\%$ train/validation split on the nodes. As usual in the inductive setting, we remove the validation nodes from the graph during training. At test time, we inductively sample 1000 times an additional node conditioned on the training graph. For each $K$, we sample 10 different training graphs.

Table 1: Mean accuracy of the Bayes classifier on test nodes $v$ with $(\mathbf{X}, \mathbf{A})$ and without $(\mathbf{X})$ structure information.

| | **Accuracy (Bayes)** | |
| --- | --- | --- |
| **K** | **X** | **(X, A)** |
| 0.1 | 50.8% | 89.7% |
| 0.5 | 59.0% | 90.3% |
| 1.0 | 68.4% | 91.7% |
| 1.5 | 76.5% | 93.1% |
| 2.0 | 83.4% | 94.7% |
| 3.0 | 92.6% | 97.4% |
| 4.0 | 97.5% | 99.0% |
| 5.0 | 99.3% | 99.8% |

**Models and attacks.** We study a wide range of popular GNN architectures: Graph Convolutional Networks (GCN) (Kipf & Welling, 2017), Simplified Graph Convolutions (SGC) (Wu et al., 2019a), Graph Attention Networks (GAT) (Veličković et al., 2018), GATv2 (Brody et al., 2022), APPNP (Gasteiger et al., 2019), and GraphSAGE (Hamilton et al., 2017). Furthermore, we study Label Propagation (LP) (Zhou et al., 2003) and a Multi-Layer Perceptron (MLP). As adversarial attacks we employ *Nettack* (Zügner et al., 2018) and *DICE* (Waniek et al., 2018) (random deletion of same-class edges and addition of different-class edges). To find over-robust examples, we use a "weak" attack we call $\ell_2$-*weak*: Connect to the closest different-class nodes in $\ell_2$-norm. Theorem 2 in Appendix B.2 shows that a strategy to change, with least structure changes, the true most likely class on CSBMs, i.e. an "optimal attack" against the Bayes classifier $g$, is given by (arbitrarily) disconnecting same-class edges and adding different-class edges to the target node. Therefore, the attacks *DICE* and $\ell_2$-*weak* have the same effect on the semantic content of a graph. We vary local budgets $\Delta$ from 1 up to the degree (deg) of a node $+2$, similarly to Zügner et al. (2018). We call the induced perturbation set $\mathcal{B}_\Delta(\cdot)$. Further details, including the hyperparameter settings can be found in Appendix E.

**Robustness metrics**. To compare the robustness across different graphs and models, we develop metrics summarizing the robustness properties of a model $f$ on a given graph. To correct for the different node degrees, we measure the adversarial robustness of $f$ (w.r.t. $g$) at node $v$ relative to $v$'s degree and average over all test nodes $V'^1$: $R(f,g) = \frac{1}{|V'|} \sum_{v \in V'} Robustness(f,g,v)/deg(v)$,

---

[1] Excluding degree 0 nodes. This is one limitation of this metric, however, these are non-existing in common benchmark datasets such as CORA and very rare in the generated CSBM graphs (Figure 8 in Appendix F.1).

where $g$ represents the Bayes classifier and hence, $Robustness(f, g, v)$ refers to the number of semantics-preserving structure changes $f$ is robust against (Definition 1). Exemplary, $R(f, g) = 0.5$ would mean that on average, node predictions are robust against changing 50% of the neighbourhood structure. Using the labels $y$ instead of a reference classifier $g$, yields the conventional (degree-corrected) adversarial robustness, unaware of semantic change, which we denote $R(f) := R(f, y)$. To measure **over-robustness**, we take the fraction of conventional adversarial robustness $R(f)$, which cannot be explained by semantic-preserving robustness $R(f, g)$: $R^{over} = 1 - R(f, g)/R(f)$. Exemplary, $R^{over} = 0.2$ means that 20% of the measured robustness is robustness beyond semantic change. In Appendix C we present a similar metric for semantic-aware adversarial robustness and how to calculate an overall robustness measure using the harmonic mean of both metrics.

## 4.1 Extent of semantic content change in common perturbation models

We investigate how prevalent perturbed graphs with changed semantic content are for common threat models. $\mathcal{B}_\Delta(\cdot)$ denotes the perturbation set allowing a local budget of $\Delta$ edge changes. Table 2 shows the fraction of test nodes, for which we find perturbed graphs in $\mathcal{B}_\Delta(\cdot)$ with changed ground truth labels, as measured by $g$. Surprisingly, even for very modest budgets, if structure matters ($K \leq 3$), this fraction is **significant**. Exemplary, for $K=1.0$ and $\mathcal{B}_{\mathrm{deg}+2}(\cdot)$, we find perturbed graphs with changed semantic content, for 99.4% of the target nodes. This establishes a *negative* answer to a question formulated in the introduction: *If structure matters, does completely reconnecting a node preserve its semantic content?* Similar to CSBMs, nodes in real-world graphs have mainly low-degree (Figure 8 in Appendix F.1). This provides evi-

Table 2: Percentage (%) of test nodes with perturbed graphs in $\mathcal{B}_\Delta(\cdot)$ violating semantic content preservation. Calculated by connecting $\Delta$ different class nodes ($\ell_2$-weak) to every target node. For $K \geq 4.0$ structure is not necessary for good generalization (Table 1). Results using DICE or Nettack are similar (see Appendix F.4). Standard deviations are insignificant and hence, omitted.

| Threat models | K | | | | | | | |
|---|---|---|---|---|---|---|---|---|
| | 0.1 | 0.5 | 1.0 | 1.5 | 2.0 | 3.0 | 4.0 | 5.0 |
| $\mathcal{B}_1(\cdot)$ | 14.3 | 11.2 | 9.1 | 6.8 | 4.4 | 1.9 | 0.8 | 0.2 |
| $\mathcal{B}_2(\cdot)$ | 35.9 | 31.2 | 25.7 | 19.8 | 14.1 | 6.2 | 2.2 | 0.7 |
| $\mathcal{B}_3(\cdot)$ | 58.5 | 53.8 | 46.8 | 38.2 | 28.8 | 14.3 | 5.1 | 1.7 |
| $\mathcal{B}_4(\cdot)$ | 76.5 | 73.0 | 66.6 | 58.1 | 47.0 | 25.7 | 9.8 | 3.4 |
| $\mathcal{B}_{\mathrm{deg}}(\cdot)$ | 75.7 | 60.0 | 55.4 | 49.1 | 39.6 | 21.9 | 9.0 | 3.2 |
| $\mathcal{B}_{\mathrm{deg}+2}(\cdot)$ | 100 | 100 | 99.4 | 92.9 | 80.5 | 51.7 | 24.8 | 9.1 |

dence that similar conclusion could be drawn for certain real-world graphs. The examined $\mathcal{B}_\Delta(\cdot)$ subsume all threat models against edge-perturbations employing the $\ell_0$-norm to measure small changes to the graph's structure, as we investigate the lowest choices of local budgets possible (see Appendix D for a discussion on global budgets). Table 2 reports lower bounds on the prevalence of graphs in $\mathcal{B}_\Delta(\cdot)$ with changed semantic content, calculated by connecting $\Delta$ different-class nodes ($\ell_2$-weak) to every target node. Thus, the perturbed graphs $\tilde{\mathcal{G}}$ are at the boundary of $\mathcal{B}_\Delta(\cdot)$. Then, we count how many $\tilde{\mathcal{G}}$ have changed the true most likely class using the Bayes classifier $g$. Exemplary, if a classifier $f$ is robust against $\mathcal{B}_3(\cdot)$ for CSBMs with $K=1.0$, $f$ classifies the found graphs at the boundary of $\mathcal{B}_3(\cdot)$ wrong in 47% of cases. Therefore, $f$ shows high **over-robustness** $R^{over}$.

## 4.2 Over-robustness of graph neural networks

Section 4.1 establishes the **existence** of perturbed graphs with changed semantic content in common threat models. Now, we examine how much of the measured robustness of common GNNs can actually be attributed to over-robustness ($R^{over}$), i.e. to robustness beyond semantic change. As a qualitative example, we study a local budget of the degree of the target nodes $\mathcal{B}_{\mathrm{deg}}(\cdot)$. Figure 3 shows the over-robustness of GNNs when attacking their classification of inductively added nodes. For $K \leq 3$ the graph structure is relevant in the prediction (see Table 1). We find that in this regime, a significant amount of the measured robustness of **all** GNNs can be attributed to **over-robustness**. Exemplary, 30.3% of the conventional adversarial robustness measurement of a GCN for $K=0.5$ turns out to be over-robustness, and similarly for other GNNs. LP achieves **lowest** $R^{over}$ while, for $K \leq 1.5$, hav-

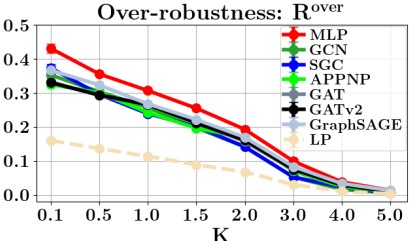

Figure 3: All GNNs show that a large share of their conventional robustness against $\mathcal{B}_{\mathrm{deg}}(\cdot)$ ($\ell_2$-weak), can be attributed to over-robustness ($R^{over}$). Nettack and DICE show similar results (Appendix F.5). $R^{over}$ is much lower for LP.

ing *best* test accuracy (Appendix F.2) and for $K \leq 1$, *best* adversarial robustness (Appendix F.6.1). Stronger attacks (e.g. Nettack) also show that a significant part of the measured conventional adversarial robustness is in fact over-robustness (see Appendix F.5). Exemplary, Nettack yields more separation between the GNNs and performs strongest against SGC and GCN. However, for a GCN at $K = 0.5$, still 11.4% of the measured robustness is in fact over-robustness. An MLP by achieving maximal adversarial provides an upper bound on the measurable over-robustness for a particular $K$. Exemplary, it has 43% over-robustness for $K = 0.1$. This means that for a perfectly robust classifier against $\mathcal{B}_{\text{deg}}(\cdot)$, 43% of the measured conventional adversarial robustness ($\ell_2$-weak) is undesirable over-robustness. Note that all GNNs are **close** to this upper bound. An MLP for Nettack for $K = 2$ still shows 19.2% $R^{over}$, indicating that here we can expect a model robust against Nettack in $\mathcal{B}_{\text{deg}}(\cdot)$ to have at least 20% undesirable over-robustness. For bounded perturbation sets $\mathcal{B}_{\Delta}(\cdot)$, maximal achievable over-robustness decreases if $K$ increases, as the more informative features are, the more structure changes it takes to change the semantic content (i.e. the Bayes decision, see Appendix F.3).

As a **positive take-away** for robustness measurements on real-world graphs, we find that if the attack is strong enough conventional adversarial robustness is a **good proxy** for semantic-aware adversarial robustness (see Appendix F.6.1). However, if the attack is weak, measured conventional robustness can turn out to be over-robustness and hence, semantic-awareness can *significantly* change robustness rankings (see Appendix F.6.2). The same holds if, using the harmonic mean, $R^{over}$ is included for an overall robustness ranking (see Appendix F.6.1). To deeper investigate the decision boundaries learned, we apply $\ell_2$-weak until it changes a model's prediction with no limited budget. This reveals that most GNNs (but not LP) have vast areas of over-robustness in input space (see Appendix F.5.4).

### 4.2.1 Over-robustness on real-world graphs

We find for CSBMs that if structure matters ($K \leq 3$), there is a large number of target nodes, which change their semantic content after only a few perturbations, mostly lower than their degree (see Appendix F.3). In contrast GNN predictions, for a majority of target nodes, are robust beyond exchanging the whole neighbourhood. This is visualized in Figure 1 measured for $K = 1.5$ and, in more detail, in Appendix F.3. On real-world graphs it is difficult to derive a reference classifier and directly measure over-robustness. However, we can investigate the degree-dependent robustness of GNNs and see if we similarly find high robustness beyond the degree of nodes. Figure 4 shows that a majority of test node predictions of a GCN on (inductive) Cora-ML (Bo-

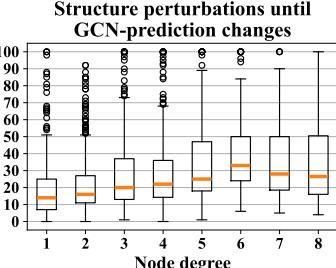

Figure 4: (Inductive) Cora-ML.

jchevski & Günnemann, 2018) are robust, by several multiples, beyond their degree. Even for degree 1 nodes, the median robustness lies at over 10 structure changes. Here, GCN's degree-dependent robustness substantially exceeds the one on CSBM-graphs with $K = 1.5$ (see Figure 11b in Appendix F.3), for which we already find high over-robustness. Results are obtained by applying a variant of $\ell_2$-weak on a target node $v$. However, as Cora-ML has multiple classes, we ensure all inserted edges connect to the same class $c'$, which is different to the class of $v$. The vulnerability of a GCN to adversarial attacks likely stems from the lower quartile of robust node classifications in Figure 4. We conjecture *over-robustness* for the upper quartile of highly robust node classifications. In Appendix F.6.3 we show similar results on Citeseer (Sen et al., 2008) and that LP, while achieving only slightly worse test accuracy, does not have such an upper quartile of highly robust node predictions.

## 5 Conclusion

We have shown that common threat models on CSBMs include a large share of graphs with changed semantic content and that GNNs show significant over-robustness to these graphs, with similar behaviour on real-world data. But we also found that the same threat models include truly adversarial examples. This dichotomy is caused by the low-degrees of nodes and their class-membership being dependent on their neighbourhood. Our results have multiple implications and we refer to Appendix H for an extended discussion. One is that for CSBMs (full) conventional robustness would lead to sub-optimal generalization and undesirable robustness *beyond* the point of semantic change. We think this calls for more caution when applying $\ell_0$-norm restricted threat models in practice and that these should not be an end to, but the beginning of an investigation into realistic graph perturbations.

## Acknowledgements

The authors want to thank Bertrand Charpentier, Yan Scholten, Marten Lienen and Nicholas Gao for valuable discussions. Furthermore, Jan Schuchardt, Filippo Guerranti, Johanna Sommer and David Lüdke for feedback to the manuscript. This paper has been supported by the DAAD programme Konrad Zuse Schools of Excellence in Artificial Intelligence, sponsored by the German Federal Ministry of Education and Research, and the German Research Foundation, grant GU 1409/4-1.

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

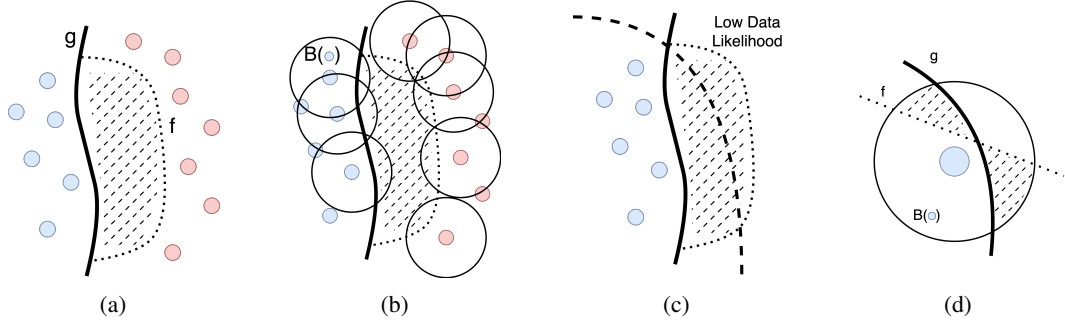

Figure 5: Conceptual differences between over- and adversarial robustness. a) The decision boundary of classifier $f$ follows the one of a base classifier $g$ except for the dotted line. b) Finite perturbation budgets induce bounded perturbation sets $\mathcal{B}(\cdot)$ intersecting only from one side with the dashed area. c) The red class is not seen because it lies in a low data likelihood region. d) Zoomed: A node whose perturbation set includes a region of adversarial and over-robust examples.

## A    Conceptual differences between over- and adversarial robustness

Figure 5a shows the decision boundary of a classifier $f$ following the one of a base classifier $g$ except for the dotted line. The dashed region between $f$'s and $g$'s decision boundary is a region of over-robustness for the blue class and a region of adversarial examples for the red class.

In the main text, we discussed the case of bounded perturbation sets (Figure 5b). What would happen, if we would lift the restriction of finite budgets? Then, for a given fixed graph and node $v$, over- and (semantic-aware) adversarial robustness are clearly distinct phenomena: Denote $\mathcal{G} = (\mathbf{X}, \mathbf{A})$ and collect all over-robust examples in a set $\mathcal{B}_O(\mathcal{G}, v) \subset \mathcal{B}(\mathcal{G})$ and adversarial examples in a set $\mathcal{B}_A(\mathcal{G}, v) \subset \mathcal{B}(\mathcal{G})$. Then, the following observation directly follows from Definition 1 and 2:

**Proposition 1.** *The set of over-robust examples $\mathcal{B}_O(\mathcal{G}, v)$ and the set of adversarial examples $\mathcal{B}_A(\mathcal{G}, v)$ are disjoint.*

However, looking at Figure 5a, one could ask if a region of over-robust examples can always be interpreted as an identical region of adversarial examples for different data points. In the following, we will show that this is *not* true in general. To come to this result, we restate the question: For a given over-robust example $\tilde{\mathcal{G}} \in \mathcal{B}(\mathcal{G})$ can we always find a corresponding clean graph $\mathcal{G}' \neq \mathcal{G}$ not in $\mathcal{B}(\mathcal{G})$ for which $\tilde{\mathcal{G}}$ is an adversarial example? To answer this, it suffices to look at the case when the node classifier $f$ is constant:

**Proposition 2.** *Given $f$ is a constant classifier, then $\mathcal{B}_A(\mathcal{G}, v)$ is empty for every possible graph $\mathcal{G}$.*

This follows as $f$ can never fulfill both, item (i) and item (iii) in Definition 1. However, over-robust examples for $f$ may exists. Every example in $\mathcal{B}(\mathcal{G})$ will be over-robust, for which $g$ changes its label for node $v$. This shows that *over-robustness* can differentiate two classifiers, which have the same adversarial robustness, but one has learned a better decision boundary, while the other has not.

Now, we come to other interesting conceptual cases. In Figure 5a it is assumed that datapoints on both sides of the decision boundary $g$ have been sampled. This may be likely if every datapoint in the input space has a comparable sampling probability. However, in practice there are regions of high and low data likelihood and hence, it could be that datapoints on one side of the dashed regions have not been sampled as exemplified in Figure 5c. There, the dashed line indicates the transition from a high to low data likelihood area. As a result, only the concept of over-robustness can capture the misbehaviour of the classifier $f$. The reverse scenario is also possible in which only the examples of the right class are sampled and the left class is in a low likelihood region. Indeed, in our results (see Section 4.2), there are some cases where we measure both high adversarial and over-robustness, exactly fitting the scenario visualized in Figure 5c. Note that it makes sense to robustify a classifier even against low-likelihood events as in safety-critical scenarios correct behaviour for unusual or rare events is crucial (Hendrycks et al., 2021).

Figure 5d zooms in to a more intricate case. Disregarding the base classifier $g$ would lead to wrongly interpreting every example above the decision boundary of $f$ as adversarial. With our refined notions,

it is possible to correctly identify the adversarial region as above $f$ until $g$, the over-robust region as below the decision boundary of $f$ but right of $g$ and the correctly classified area in the top-right. Our results show that this case is prevalent for GNNs using common threat models, as if structure matters, we find over-robust examples (see Section 4.2 and Appendix F.3) as well as adversarial examples (see Appendix F.6.1) in the examined perturbation sets.

## B  Proofs

### B.1  Bayes optimal classifier

We proof Theorem 1 by deriving the Bayes classifier for multi-class node classification with $C$ classes. Theorem 1 then follows as special case by setting $C = 2$. We restate Theorem 1 for multiple classes:

**Theorem 1.** *The Bayes optimal classifier, minimizing the expected* $0/1$*-loss (1), is* $f^*(\mathbf{X}', \mathbf{A}', y)_v = \arg\max_{\hat{y} \in \{0,...,C-1\}} \mathbb{P}[y'_v = \hat{y} | \mathbf{X}', \mathbf{A}', y]$.

*Proof.* Lets denote by $x_{-i}$ all elements of a vector $x$ except the $i$-th one. First, note that $\mathcal{D}(\mathbf{X}, \mathbf{A}, y)$ from which we sample $(\mathbf{X}', \mathbf{A}', y')$ defines a conditional joint distribution

$$
\begin{aligned}
\mathbb{P}[\mathbf{X}', \mathbf{A}', y' | \mathbf{X}, \mathbf{A}, y] &= \mathbb{P}[y'_v | \mathbf{X}', \mathbf{A}', y'_{-v}, \mathbf{X}, \mathbf{A}, y] \cdot \mathbb{P}[\mathbf{X}', \mathbf{A}', y'_{-v} | \mathbf{X}, \mathbf{A}, y] \\
&= \mathbb{P}[y'_v | \mathbf{X}', \mathbf{A}', y] \cdot \mathbb{P}[\mathbf{X}', \mathbf{A}', y | \mathbf{X}, \mathbf{A}, y] \\
&= \mathbb{P}[y'_v | \mathbf{X}', \mathbf{A}', y] \cdot \mathbb{P}[\mathbf{X}', \mathbf{A}' | \mathbf{X}, \mathbf{A}, y]
\end{aligned}
\tag{2}
$$

where the first line follows from the basic definition of conditional probability, the second lines from the definition of the inductive sampling scheme (i.e., $y'_{-v} = y$ and similar for $\mathbf{X}'$ and $\mathbf{A}'$), and the third line from $\mathbb{P}[y | \mathbf{X}, \mathbf{A}, y] = 1$. Now, we can rewrite the expected loss (1) with respect to these probabilities as

$$
\begin{aligned}
&\mathbb{E}_{\mathbf{X}', \mathbf{A}' | \mathbf{X}, \mathbf{A}, y} \left[ \mathbb{E}_{y'_v | \mathbf{X}', \mathbf{A}', y} [\ell_{0/1}(y'_v, f(\mathbf{X}', \mathbf{A}', y)_v)] \right] \\
&= \mathbb{E}_{\mathbf{X}', \mathbf{A}' | \mathbf{X}, \mathbf{A}, y} \left[ \sum_{k=0}^{C-1} \ell_{0/1}(k, f(\mathbf{X}', \mathbf{A}', y)_v) \cdot \mathbb{P}[y'_v = k | \mathbf{X}', \mathbf{A}', y] \right]
\end{aligned}
\tag{3}
$$

The following argument is adapted from Hastie et al. (2009). Equation (3) is minimal for a classifier $f^*$, if it is (point-wise) minimal for every $(\mathbf{X}', \mathbf{A}', y)$. This means

$$
\begin{aligned}
f^*(\mathbf{X}', \mathbf{A}', y)_v &= \arg\min_{\hat{y} \in \{0,...,C-1\}} \sum_{k=0}^{C-1} \ell_{0/1}(k, \hat{y}) \cdot \mathbb{P}[y'_v = k | \mathbf{X}', \mathbf{A}', y] \\
&= \arg\min_{\hat{y} \in \{0,...,C-1\}} \sum_{k=0}^{C-1} (1 - \mathbb{I}[k = \hat{y}]) \cdot \mathbb{P}[y'_v = k | \mathbf{X}', \mathbf{A}', y] \\
&= \arg\max_{\hat{y} \in \{0,...,C-1\}} \sum_{k=0}^{C-1} \mathbb{I}[k = \hat{y}] \cdot \mathbb{P}[y'_v = k | \mathbf{X}', \mathbf{A}', y] \tag{4} \\
&= \arg\max_{\hat{y} \in \{0,...,C-1\}} \mathbb{P}[y'_v = \hat{y} | \mathbf{X}', \mathbf{A}', y] \tag{5}
\end{aligned}
$$

where line (5) follows from fact that in the sum of line (4), there can only be one non-zero term. Equation (5) tells us that the optimal decision is to choose the most likely class. Due to $f^*(\mathbf{X}', \mathbf{A}', y) = \arg\max_{\hat{y} \in \{0,...,C-1\}} \mathbb{P}[y'_v = \hat{y} | \mathbf{X}', \mathbf{A}', y]$ minimizing (3) it also minimizes the expected loss (1) and hence, is a Bayes optimal classifier. $\qquad\square$

## B.2   Optimal attack on CSBMs

**Theorem 2.** *Given a graph generated by a CSBM. The minimal number of structure changes to change the Bayes classifier (Theorem 1) for a target node $v$ is defined by iteratively: i) connecting $v$ to another node $u$ with $y_v \neq y_u$ or ii) dropping a connection to another node $u$ with $y_v = y_u$.*

*Proof.* Assume $(\mathbf{X}', \mathbf{A}', y') \sim \text{CSBM}_{1,p,q}^{\mu,\sigma^2}(\mathbf{X}, \mathbf{A}, y)$ with $q < p$ (homophily assumption). Recall the Bayes decision $y^* = \arg\max_{\hat{y} \in \{0,1\}} \mathbb{P}[y'_v = \hat{y} | \mathbf{X}', \mathbf{A}', y]$. We want to prove which structure perturbations result in a minimally changed adjacency matrix $\tilde{\mathbf{A}}'$, as measured using the $\ell_0$-norm, but for which

$$y_{new}^* = \arg\max_{\hat{y} \in \{0,1\}} \mathbb{P}[y'_v = \hat{y} | \mathbf{X}', \tilde{\mathbf{A}}', y] \neq y^* \tag{6}$$

Therefore, we want to change

$$\mathbb{P}[y'_v = y^* | \mathbf{X}', \mathbf{A}', y'] > \mathbb{P}[y'_v = 1 - y^* | \mathbf{X}', \mathbf{A}', y'] \tag{7}$$

to

$$\mathbb{P}[y'_v = y^* | \mathbf{X}', \tilde{\mathbf{A}}', y'] < \mathbb{P}[y'_v = 1 - y^* | \mathbf{X}', \tilde{\mathbf{A}}', y'] \tag{8}$$

To achieve this, first note that we can rewrite Equation (7) using Bayes theorem:

$$\frac{\mathbb{P}[\mathbf{X}'_{v,:}, \mathbf{A}'_{v,:} | y'_v = y^*, \mathbf{X}, \mathbf{A}, y] \cdot \mathbb{P}[y'_v = y^* | \mathbf{X}, \mathbf{A}, y]}{\mathbb{P}[\mathbf{X}'_{v,:}, \mathbf{A}'_{v,:} | \mathbf{X}, \mathbf{A}, y]}$$
$$> \frac{\mathbb{P}[\mathbf{X}'_{v,:}, \mathbf{A}'_{v,:} | y'_v = 1 - y^*, \mathbf{X}, \mathbf{A}, y] \cdot \mathbb{P}[y'_v = 1 - y^* | \mathbf{X}, \mathbf{A}, y]}{\mathbb{P}[\mathbf{X}'_{v,:}, \mathbf{A}'_{v,:} | \mathbf{X}, \mathbf{A}, y]} \tag{9}$$
$$\iff \mathbb{P}[\mathbf{X}'_{v,:}, \mathbf{A}'_{v,:} | y'_v = y^*, \mathbf{X}, \mathbf{A}, y] > \mathbb{P}[\mathbf{X}'_{v,:}, \mathbf{A}'_{v,:} | y'_v = 1 - y^*, \mathbf{X}, \mathbf{A}, y] \tag{10}$$

where in Equation 9 we use $(\mathbf{A}'_{v,:})^T = \mathbf{A}'_{:,v}$ and Equation 10 follows from $\mathbb{P}[y'_v = y^* | \mathbf{X}, \mathbf{A}, y] = \mathbb{P}[y'_v = 1 - y^* | \mathbf{X}, \mathbf{A}, y] = \frac{1}{2}$ by definition of the sampling process. Now, we take the logarithm of both sides in (10) and call the log-difference $\Delta$:

$$\Delta(\mathbf{A}') := \log \mathbb{P}[\mathbf{X}'_{v,:}, \mathbf{A}'_{v,:} | y'_v = y^*, \mathbf{X}, \mathbf{A}, y] - \log \mathbb{P}[\mathbf{X}'_{v,:}, \mathbf{A}'_{v,:} | y'_v = 1 - y^*, \mathbf{X}, \mathbf{A}, y] \tag{11}$$

Clearly, Equation 10 is equivalent to

$$\Delta(\mathbf{A}') \geq 0 \tag{12}$$

Using the properties of the sampling process of a CSBM (see Section 2), we can rewrite

$$\mathbb{P}[\mathbf{X}'_{v,:}, \mathbf{A}'_{v,:} | y'_v = y^*, \mathbf{X}, \mathbf{A}, y] = \mathbb{P}[\mathbf{X}'_{v,:} | y'_v = y^*] \cdot \mathbb{P}[\mathbf{A}'_{v,:} | y'_v = y^*, y] \tag{13}$$
$$= \mathbb{P}[\mathbf{X}'_{v,:} | y'_v = y^*] \cdot \prod_{i \in [n] \setminus \{v\}} \mathbb{P}[\mathbf{A}'_{v,i} | y'_v = y^*, y_i] \tag{14}$$

and therefore

$$\Delta(\mathbf{A}') = \log \frac{\mathbb{P}[\mathbf{X}'_{v,:} | y'_v = y^*]}{\mathbb{P}[\mathbf{X}'_{v,:} | y'_v = 1 - y^*]}$$
$$+ \sum_{i \in [n] \setminus \{v\}} \underbrace{\left( \log \mathbb{P}[\mathbf{A}'_{v,i} | y'_v = y^*, y_i] - \log \mathbb{P}[\mathbf{A}'_{v,i} | y'_v = 1 - y^*, y_i] \right)}_{\Delta_i(\mathbf{A}')} \tag{15}$$

Now, to achieve (8), we want to find those structure perturbations, which lead to $\Delta(\tilde{\mathbf{A}}') < 0$ the fastest (i.e., with least changes). First, note that the first term in Equation 15 does not depend on the adjacency matrix and hence, can be ignored. The second term shows that the change in $\Delta(\mathbf{A}')$ induced by adding or removing an edges $(v, i)$ is additive and independent of adding or removing another edge $(v, j)$. Denote by $\tilde{\mathbf{A}}'(u)$ the adjacency matrix constructed by removing (adding) edge $(v, u)$ from $\mathbf{A}'$ if $(v, u)$ is (not) already in the graph. We define the change potential of node as $u$ as $\tilde{\Delta}_u := \Delta_u(\tilde{\mathbf{A}}'(u)) - \Delta_u(\mathbf{A}')$. Then, we only need find those nodes $u$ with maximal change potential $|\tilde{\Delta}_u| = |\Delta_u(\tilde{\mathbf{A}}'(u)) - \Delta_u(\mathbf{A}')|$ and $\tilde{\Delta}_u < 0$ and disconnect (connect) them in decreasing order of $|\tilde{\Delta}_u|$ until $\Delta(\tilde{\mathbf{A}}') < 0$. We will now show that any node $u$ has maximal negative change potential, who either satisfies i) $y_u = y_*$ and $\mathbf{A}'_{v,u} = 1$ or ii) $y_u \neq y_*$ and $\mathbf{A}'_{v,u} = 0$.

To prove this, we make a case distinction on the existence of $(v, u)$ in the unperturbed graph and the class of $y_u$:

**Case $\mathbf{A}'_{v,u} = 0$:**

We distinguish two subcases:

i) $y_u \neq y_*$:

We can write

$$
\begin{aligned}
\tilde{\Delta}_u &= \log \mathbb{P}[\mathbf{A}'_{v,i} = 1 | y'_v = y^*, y_i] - \log \mathbb{P}[\mathbf{A}'_{v,i} = 1 | y'_v = 1 - y^*, y_i] \\
&\quad - \log \mathbb{P}[\mathbf{A}'_{v,i} = 0 | y'_v = y^*, y_i] + \log \mathbb{P}[\mathbf{A}'_{v,i} = 0 | y'_v = 1 - y^*, y_i] \\
&= \log q - \log p - \log(1 - q) + \log(1 - p) \quad &(16) \\
&< 0 \quad &(17)
\end{aligned}
$$

Equation 16 follows from the sampling process of the CSBM. Equation 17 follows from $q < p$, implying $\log q - \log p < 0$ and $-\log(1 - q) + \log(1 - p) < 0$.

ii) $y_u = y_*$

We can write

$$
\tilde{\Delta}_u = \log p - \log q - \log(1 - p) + \log(1 - q) > 0 \quad (18)
$$

where the last $>$ follows similarly from $q < p$.

**Case $\mathbf{A}'_{v,u} = 1$:**

i) $y_u \neq y_*$:

We can write

$$
\begin{aligned}
\tilde{\Delta}_u &= \log \mathbb{P}[\mathbf{A}'_{v,i} = 0 | y'_v = y^*, y_i] - \log \mathbb{P}[\mathbf{A}'_{v,i} = 0 | y'_v = 1 - y^*, y_i] \\
&\quad - \log \mathbb{P}[\mathbf{A}'_{v,i} = 1 | y'_v = y^*, y_i] + \log \mathbb{P}[\mathbf{A}'_{v,i} = 1 | y'_v = 1 - y^*, y_i] \\
&> 0 \quad &(19)
\end{aligned}
$$

where Equation 19 follows by the insight, that $\tilde{\Delta}_u$ is the same as for case $\mathbf{A}'_{v,u} = 0$ except multiplied with $-1$.

ii) $y_u = y_*$

We can write

$$
\tilde{\Delta}_u = \log q - \log p - \log(1 - q) + \log(1 - p) < 0 \quad (20)
$$

where the first equality follows from the insight, that $\tilde{\Delta}_u$ is again the same as for case $\mathbf{A}'_{v,u} = 0$ except multiplied with $-1$. The last $>$ follows again from $q < p$.

The theorem follows from the fact that only the cases where we add an edge to a node of different class, or drop an edge to a node with the same class have negative change potential and the fact, that both cases have the same change potential.

$\square$

## C   Robustness metrics

We restate the degree corrected robustness of a classifier $f$ w.r.t. a reference classifier $g$:

$$R(f,g) = \frac{1}{|V'|} \sum_{v \in V'} \frac{\text{Robustness}(f,g,v)}{\deg(v)} \tag{21}$$

Using the true labels $y$ instead of a reference classifier $g$ in (21), one can measure the maximal achievable robustness (before semantic content changes) as $R(g) := R(g,y)$, i.e. the semantic boundary. Note that we can exactly compute $R(g)$ due to knowledge of the data generating process. We measure **adversarial robustness** as the fraction of optimal robustness $R(g)$ achieved: $R^{adv} = R(f,g)/R(g)$. Again, to correctly measure $R^{adv}$, the identical attack is performed to measure $R(f,g)$ and $R(g)$.

A model $f$ can have high adversarial- but also high over-robustness (see Section 3). To have a metric, which truly shows a complete picture of the robustness properties of a model, we take the harmonic mean of $R^{adv}$ and the percentage of how much robustness is legitimate $(1 - R^{over})$ and define an **$F_1$-robustness score**: $F_1^{rob}(\cdot,\cdot) = 2\frac{(1-R^{over})\cdot R^{adv}}{(1-R^{over})+R^{adv}}$. Only a model showing perfect adversarial robustness and no over-robustness achieves $F_1^{rob} = 1$.

Note that using $F_\beta^{rob} = (1 + \beta^2)\frac{(1-R^{over})\cdot R^{adv}}{\beta^2(1-R^{over})+R^{adv}}$, with $\beta \geq 0$, one can weight the importance of over- versus adversarial examples with $R^{adv}$ being $\beta$-times as important as $(1 - R^{over})$.

## D   Global budget restrictions

Global budgets again result in local edge-perturbations, however, now commonly allowing for way stronger perturbations than $\mathcal{B}_{\deg+2}(\cdot)$ to individual nodes. The number of allowed perturbation is usually set to a small one or two digit percentage of the total number of edges in the graph. For CORA with 5278 edges, a relative budget of 5% leads to a budget of 263 edge-changes, which can be distributed in the graph without restriction.

## E   Experiment details

**Dataset.**

We use Contextual Stochastic Block Models (CSBMs). The main setup is described in Section 4. Table 3 summarizes some dataset statistics and contrasts them with CORA. The reported values are independent of $K$, hence we report the average across 10 sampled CSBM graphs for one $K$.

Table 3: Dataset statistics.

| Dataset | # Nodes | # Edges | # Features | # Classes | Average node degree | Average same-class node degree | Average different-class node degree |
|---------|---------|---------|------------|-----------|---------------------|--------------------------------|------------------------------------|
| CSBM | 1,000 | 1,964±25 | 21 | 2 | 3.92±0.05 | 3.25±0.05 | 0.67±0.02 |
| CORA | 2,708 | 5,278 | 1,433 | 7 | 3.90 | 3.16 | 0.74 |

Figure 6a shows that CSBM graphs mainly contain low-degree nodes.

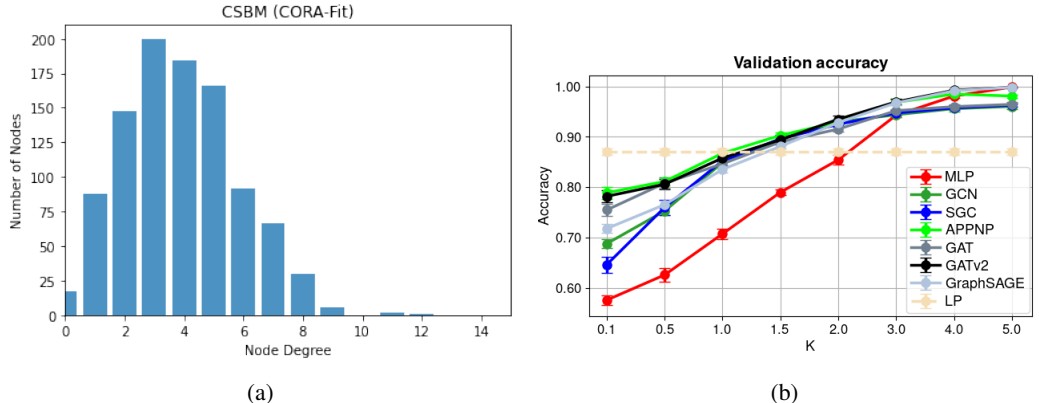

(a)             (b)

Figure 6: (a) Degree distribution of a CSBM graph as parametrized in Section 4 ($n = 1000$). (b) Validation accuracies of the best performing hyperparameters of the different models. Note that GNNs for low $K$ (high-structure relevance) underperform pure LP.

**Graph neural networks and label propagation (LP).**

We perform extensive hyperparameter search for each GNN model, LP and MLP for each individual $K$ and choose, for each $K$, the on average best performing hyperparamters on 10 graphs sampled from the respective CSBM. Interestingly, we find that very different hyperparameters are optimal for different choices of the feature-information defining parameter $K$. We also find that using the default parameters from the respective model papers successful on the benchmark real-world datasets, don't work well for some choices of $K$, especially low $K$ when structure is very important but features not so.

We train all model for 3000 epochs with a patients of 300 epochs using Adam (Kingma & Ba, 2015) and explore learning rates $[0.1, 0.01, 0.001]$ and weight decay $[0.01, 0.001, 0.001]$ and additionally for

- *MLP:* We use a 1 (Hidden)-Layer MLP and test hidden dimensions $[32, 64, 128, 256]$ and dropout $[0.0, 0.3, 0.5]$. We employ the ReLU activation function.

- *LP:* We use label spreading (Zhou et al., 2003) as basic label propagation technique. We use 50 iterations and test $\alpha$ in the range between $0.00$ and $1.00$ in step sizes of $0.05$. LP is the only method having the same hyperparameters on all CSBMs, as it is independent of $K$.

- *SGC:* We explore $[1, 2, 3, 4, 5]$ number of hops and additionally, a learning rate of $0.2$. We investigate dropouts of $[0, 0.3, 0.5]$. SGC was the most challening to train for low $K$.

- *GCN:* We use a two layer (ReLU) GCN with $64$ filters and dropout $[0.0, 0.3, 0.5]$

- *GAT:* We use a two layer GAT with 8 heads and 8 features per head with LeakyReLU having a negative slope of $0.2$. We test dropout $[0.0, 0.3, 0.6]$ and neighbourhood dropout $[0.0, 0.3, 0.6]$.

- *GATv2:* We use the best performing hyperparameters of GAT.

- *APPNP:* We use 64 hidden layers, $K = 10$ iterations, dropout $[0.0, 0.3, 0.6]$ and $[0.0, 0.3, 0.5]$ and test $\alpha$ in $[0.05, 0.1, 0.2]$. Interestingly, the higher $K$, we observe higher $\alpha$ performing better.

The (averaged) validation accuracies can be seen in Figure 6b.

**Attacks.**

- *Nettack* attacks a surrogate SGC model to approximately find maximal adversarial edges. As a surrogate model, instead of using a direct SGC implementation as in the models section, we use a 2-Layer GCN with the identity function as non-linearity and 64 filters and found it trains easier on $K = 0.1$ and hence, provides better adversarial examples for $K = 01$ than direct SGC implementation. For higher $K$, differences are neglectable. We use the same

hyperparamter search as outlined for the conventional GCN. As CSBMs do not follow a power-law in their degree distribution, we removed this distributional test from Nettack, making it effectively stronger.

- *DICE* randomly disconnects $d$ edges from the test node $v$ to same-class nodes and connects $b$ edges from $v$ to different-class nodes. For a given local budget $\Delta$, we set $d = 0$ and $b = \Delta$.

### E.1 Real-world graphs

We introducing the experimental setup for evaluating overrobustness of GNNs on real-world graphs in detail. This includes datasets, models and evaluation procedure.

**Datasets.** To explore overrobustness on real-world graphs, the citation networks Cora-ML (Bojchevski & Günnemann, 2018) and Citeseer (Sen et al., 2008) are selected. Table 4 provides an overview over the most important dataset characteristics. Figure 7 visualizes the degree distributions up to degree 15. For both datasets, 40 nodes per class are randomly selected as validation and test nodes. The remaining nodes are selected as labeled training set. On Cora-ML, this results in approximately 80% training, 10% validation and 10% test nodes. Following the inductive approach used in Section 4 for CSBMs, model optimization is performed using the subgraph spanned by all training nodes. Early stopping uses the subgraph spanned by all training and validation nodes.

Table 4: Dataset statistics.

| **Dataset** | # Nodes | # Edges | # Features | # Classes | Average node degree | Average same-class node degree | Average different-class node degree |
|---|---|---|---|---|---|---|---|
| Cora-ML | 2,810 | 15,962 | 2,879 | 7 | 5.68 | 4.46 | 1.22 |
| Citeseer | 2,110 | 7,336 | 3,703 | 6 | 3.48 | 2.56 | 0.92 |

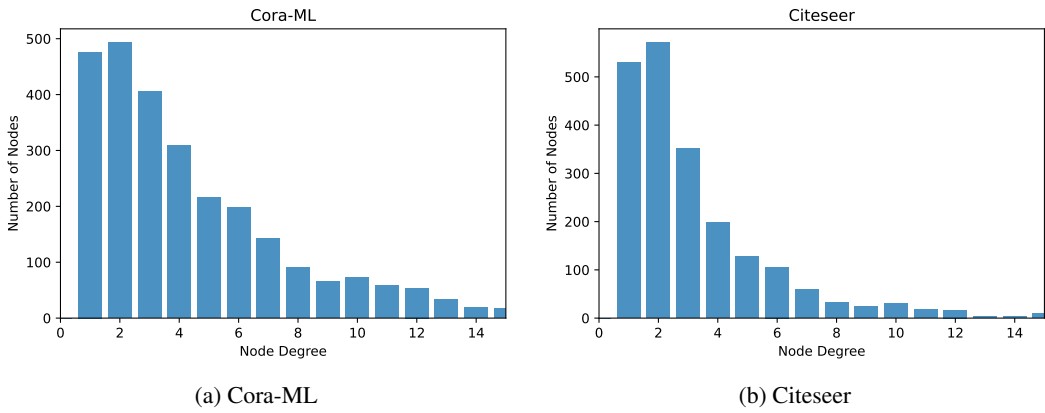

(a) Cora-ML

(b) Citeseer

Figure 7: Degree distribution by dataset.

**Model Architectures.** We evaluate the robustness of Graph Convolutional Networks (GCN), Label Propagation (LP), and GCN followed by LP post-processing (GCN+LP). The GCN architecture and optimization scheme follow Geisler et al. (2021). The GCN has two layers 64 filters. During training, a dropout of $0.5$ is applied. We optimize the model parameters for a maximum of 3000 epochs using Adam (Kingma & Ba, 2015) with learning rate $0.01$ and weight decay $0.001$. LP uses the normalized adjacency as transition matrix and is always performed for ten iterations. This mirrors related architectures like APPNP (Gasteiger et al., 2019). We additionally choose $\alpha = 0.7$ on both datasets by grid-search over $\{0.1, 0.3, 0.5, 0.7, 0.9\}$. For GCN+LP, it should be noted that LP is applied as a post-processing routine at test-time only. It is not included during training.

**Evaluating degree-depending robustness.** We investigate degree-dependent robustness of GNNs by following a similar strategy as the $\ell_2$-weak attack on CSBMs. However, real-world datasets are multi-class. Therefore, we ensure to only connect the target nodes to nodes of one selected class. More concretely, let $v$ be a correctly classified test node with label $c^*$. We investigate for each class $c \neq c^*$, how many edges we can connect to $v$, until the model's prediction changes and denote this

number $N_c(v)$. The attack then works as follows: First, we project the high-dimesional feature vectors into lower dimensional space by applying the first weight matrix of the GCN. Then, we iteratively add edges connecting $v$ to the most similar nodes (after projection) in $\ell_2$-norm from $c$ and evaluate after how many insertions the model's prediction changes. We present the results for the class achieving lowest robustness, i.e. smallest $N_c(v)$, and the results for the class achieving highest robustness, i.e. largest $N_c(v)$.

# F  Further results

## F.1  Degree distribution CSBM vs CORA

Note that degrees in CSBM do **not** follow a power-law distribution. However, they are similar in a different sense to common benchmark citation networks. The goal of this section is to show that the large majority of nodes in both graphs have degrees 2, 3, 4 or 5. Low degree nodes are even more pronounced in CORA than CSBMs.

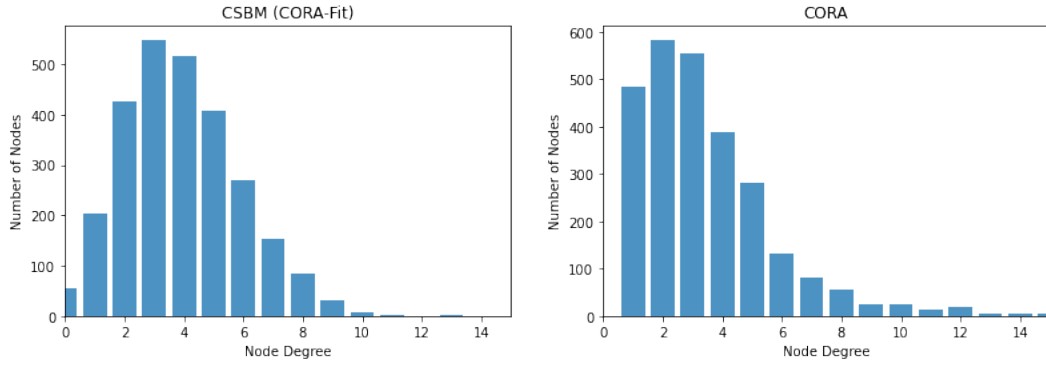

(a) Graph sampled from a CSBM, using $n = 2708$ as CORA. Note that the graph structure is of a CSBM is independent of the $K$ but only dependent on $p, q$ which have been set to fit CORA. Plot cut at node degree 15.

(b) CORA contains mainly low degree nodes. Plot cut at node degree 15.

Figure 8: Degree distribution of the used CSBMs vs CORA, both distribution show that the graphs mainly contain low-degree nodes. This is even more pronounced in CORA than CSBMs.

## F.2  Test accuracy on CSBM

This section summaries the detailed performance of the different models on the CSBMs.

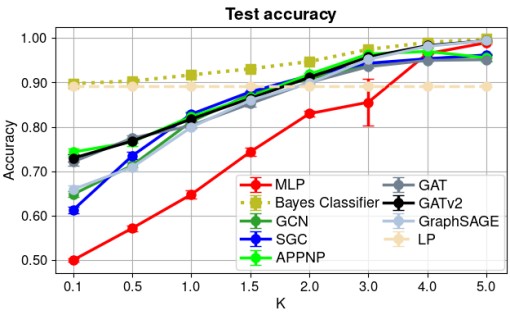

Figure 9: Test accuracy of models on test nodes on the CSBMs.

Table 5: Average test accuracies of the models on the sampled test nodes on the CSBMs.

| K | 0.1 | 0.5 | 1.0 | 1.5 | 2.0 | 3.0 | 4.0 | 5.0 |
|---|---|---|---|---|---|---|---|---|
| Bayes classifier (BC) | **89.7%** | **90.3%** | **91.7%** | **93.1%** | **94.7%** | **97.4%** | **99.0%** | **99.8%** |
| BC (Features Only) | 50.8% | 59.0% | 68.4% | 76.5% | 83.4% | 92.6% | 97.5% | 99.3% |
| BC (Structure Only) | 89.8% | 89.8% | 89.8% | 89.8% | 89.8% | 89.8% | 89.8% | 89.8% |
| MLP | 50.4% | 57.2% | 64.6% | 74.3% | 83.0% | 85.3% | 96.6% | 99.0% |
| GCN | 64.6% | 71.5% | 81.2% | 87.3% | 90.8% | 94.0% | 95.3% | 96.0% |
| SGC | 61.3% | 73.5% | 82.9% | 87.9% | 91.5% | 94.3% | 95.3% | 96.2% |
| APPNP | 74.6% | 76.7% | 82.4% | 87.7% | **91.8%** | **96.4%** | 97.0% | 95.5% |
| GAT | 70.0% | 77.6% | 80.8% | 85.2% | 89.6% | 93.7% | 95.0% | 95.5% |
| GATv2 | 72.8% | 76.7% | 81.5% | 86.1% | 90.7% | 95.3% | **98.1%** | 99.3% |
| GraphSAGE | 66.2% | 70.6% | 79.9% | 86.0% | 89.8% | 95.4% | 97.9% | **99.4%** |
| LP | **89.2%** | **89.2%** | **89.2%** | **89.2%** | 89.2% | 89.2% | 89.2% | 89.2% |

Table 6: Standard deviation of test accuracy of the models on the sampled test nodes on the CSBMs.

| K | 0.1 | 0.5 | 1.0 | 1.5 | 2.0 | 3.0 | 4.0 | 5.0 |
|---|---|---|---|---|---|---|---|---|
| Bayes classifier (BC) | 0.2% | 0.2% | 0.1% | 0.1% | 0.1% | 0.2% | 0.1% | 0.1% |
| BC (Features Only) | 1.5% | 1.5% | 1.7% | 1.8% | 1.6% | 1.1% | 0.6% | 0.3% |
| BC (Structure Only) | 0.8% | 0.8% | 0.8% | 0.8% | 0.8% | 0.8% | 0.8% | 0.8% |
| MLP | 0.5% | 0.6% | 0.9% | 0.9% | 0.6% | 5.2% | 0.3% | 0.1% |
| GCN | 0.5% | 0.5% | 0.5% | 0.3% | 0.2% | 0.2% | 0.2% | 0.1% |
| SGC | 0.8% | 0.8% | 0.4% | 0.2% | 0.1% | 0.2% | 0.1% | 0.1% |
| APPNP | 0.9% | 0.8% | 1.2% | 0.7% | 0.3% | 0.2% | 0.7% | 0.9% |
| GAT | 1.5% | 0.6% | 1.1% | 1.1% | 0.4% | 0.3% | 0.3% | 0.4% |
| GATv2 | 0.8% | 0.4% | 0.4% | 0.5% | 0.3% | 0.5% | 0.2% | 0.1% |
| GraphSAGE | 0.8% | 0.5% | 0.3% | 0.3% | 0.2% | 0.1% | 0.2% | 0.1% |
| LP | 0.3% | 0.3% | 0.3% | 0.3% | 0.3% | 0.3% | 0.3% | 0.3% |

## F.3 Average robustness of the Bayes classifier and the empirical cause of over-robustness

Figure F.3 shows that the average robustness of the Bayes classifier, increases with $K$ and is linear in the degree of the node. Furthermore, if structure matters ($K \leq 3$), the (average) robustness of the Bayes classifier rarely exceeds the degree of a node. Figure 11 shows a more detailed picture for $K = 1.5$ and contrasts the number of structure changes until the Bayes classifier changes (Figure 11a) with that of a GCN (Figure 11b). Over-robustness is caused by structure perturbations to a target node, allowed by a threat model $\mathcal{B}_\Delta(\cdot)$, for which the Bayes decision changes before the GNN decision changes. Figure 11 shows that there is a large portion of nodes, where the GCN is significantly more robust than the Bayes classifier. This strongly suggests, together with the robustness of the Bayes decision's median being at most the degree of the node, that for any local budget $\Delta$, we will find test nodes with $\leq \Delta$ edge perturbations, where the Bayes decision changes before the GCN prediction changes. Indeed, this is true for a significant fraction of nodes for any $\Delta$ as presented in Table 7. Robustness values for the Bayes classifier, i.e. the number of structure changes until semantic content changes, are calculated using an "optimal" attack strategy outlined in Theorem 2 from Appendix B.2.

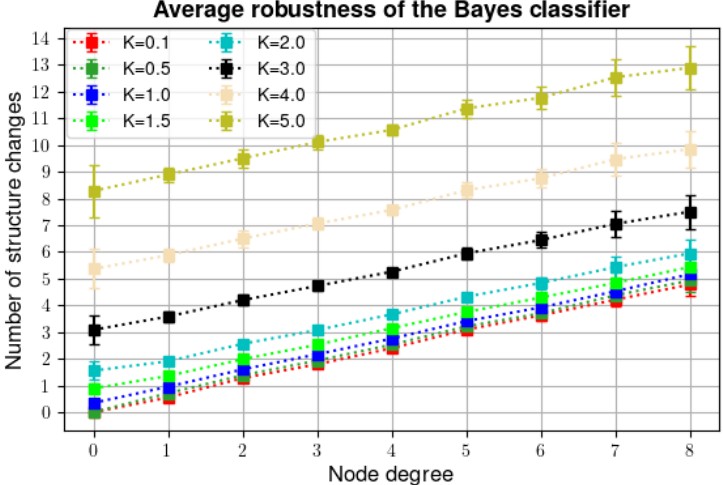

Figure 10: Average robustness of the Bayes classifier on CSBMs. Robustness values are calculated using an "optimal" attack strategy outlined in Theorem 2 from Appendix B.2.

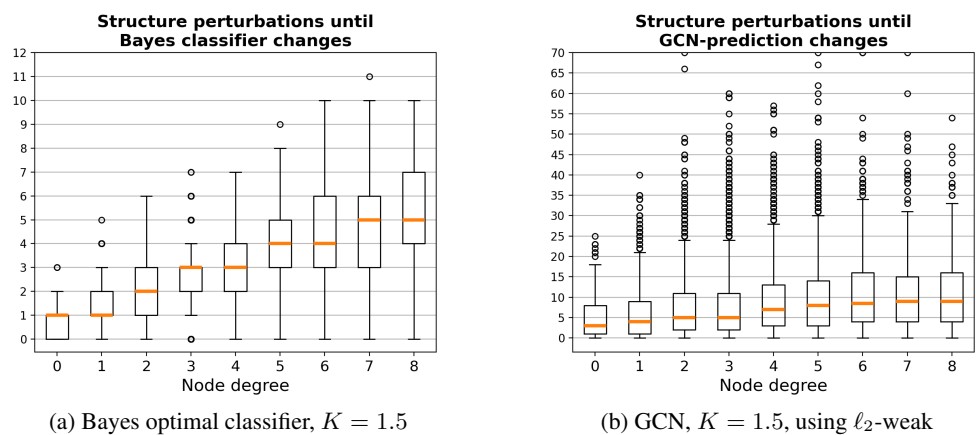

(a) Bayes optimal classifier, $K = 1.5$   (b) GCN, $K = 1.5$, using $\ell_2$-weak

Figure 11: Structure perturbations until (a) Bayes optimal classifier or (b) GCN changes for $K = 1.5$ (as in Figure 1). Data collected over 10 sampled graphs from a CSBM with $K = 1.5$. The qualitative picture is similar for other $K$, with slightly less robustness for $K < 1.5$ and more for $K > 1.5$. Robustness values for the Bayes decision are calculated using an "optimal" attack strategy outlined in Theorem 2 from Appendix B.2. For the GCN, $\ell_2$-weak is used.

Table 7: Percentage of test nodes for $K = 1.5$, for which a GCN is more robust than the Bayes decision for perturbations from a threat model $\mathcal{B}_\Delta(\cdot)$ (using $\ell_2$-weak for attacking the GCN). These nodes are responsible for the measured over-robustness, as each node is an over-robust example for the GCN in $\mathcal{B}_\Delta(\cdot)$. Qualitative picture similar for other $K$ and architectures.

| Threat models | GCN |
|---|---|
| $\mathcal{B}_1(\cdot)$ | 4.73% |
| $\mathcal{B}_2(\cdot)$ | 14.45% |
| $\mathcal{B}_3(\cdot)$ | 27.86% |
| $\mathcal{B}_4(\cdot)$ | 42.42% |
| $\mathcal{B}_{\text{deg}}(\cdot)$ | 36.41% |
| $\mathcal{B}_{\text{deg}+2}(\cdot)$ | 67.93% |

### F.4 Extent of semantic content change in common perturbation models

The extent of semantic content change looks similar for DICE and Nettack to Table 2 ($\ell_2$-weak).

**Nettack:**

Note that for $K = 0.1$ the surrogate model SGC uses by Nettack has only mediocre test-accuracy due to features not being very informative. Therefore, Nettack sometimes proposes to add same-class edges or remove different-class edges.

Table 8: Percentage (%) of nodes for which we find perturbed graphs in $\mathcal{B}_\Delta(\cdot)$ violating semantic content preservation, i.e. with changed ground truth labels. Calculated by adding or dropping $\Delta$ edges suggested by Nettack to every target node. Note that for $K=4.0$ and $K=5.0$ structure is not necessary for good generalization (Table 1). Standard deviation are insignificant and hence, omitted.

| Threat models | $K$=0.1 | $K$=0.5 | $K$=1.0 | $K$=1.5 | $K$=2.0 | $K$=3.0 | $K$=4.0 | $K$=5.0 |
|---|---|---|---|---|---|---|---|---|
| $\mathcal{B}_1(\cdot)$ | 10.6 | 9.7 | 9.1 | 6.8 | 4.4 | 1.9 | 0.7 | 0.2 |
| $\mathcal{B}_2(\cdot)$ | 23.5 | 24.7 | 24.8 | 19.8 | 14.1 | 6.2 | 2.2 | 0.7 |
| $\mathcal{B}_3(\cdot)$ | 35.7 | 41.0 | 43.6 | 38.1 | 28.8 | 14.2 | 4.9 | 1.6 |
| $\mathcal{B}_4(\cdot)$ | 47.3 | 55.0 | 61.2 | 57.5 | 46.8 | 25.1 | 9.2 | 3.2 |
| $\mathcal{B}_{\text{deg}}(\cdot)$ | 45.4 | 42.9 | 50.0 | 47.6 | 39.4 | 21.9 | 8.9 | 3.1 |
| $\mathcal{B}_{\text{deg+2}}(\cdot)$ | 63.9 | 73.7 | 90.1 | 89.8 | 79.6 | 50.2 | 23.6 | 8.6 |

**DICE:**

Table 9: Percentage (%) of nodes for which we find perturbed graphs in $\mathcal{B}_\Delta(\cdot)$ violating semantic content preservation, i.e. with changed ground truth labels. Calculated by randomly connecting $\Delta$ different-class nodes (DICE) to every target node. Note that for $K=4.0$ and $K=5.0$ structure is not necessary for good generalization (Table 1). Standard deviation are insignificant and hence, omitted.

| Threat models | $K$=0.1 | $K$=0.5 | $K$=1.0 | $K$=1.5 | $K$=2.0 | $K$=3.0 | $K$=4.0 | $K$=5.0 |
|---|---|---|---|---|---|---|---|---|
| $\mathcal{B}_1(\cdot)$ | 14.9 | 10.9 | 9.0 | 6.2 | 4.3 | 2.1 | 0.6 | 0.2 |
| $\mathcal{B}_2(\cdot)$ | 36.8 | 31.4 | 26.2 | 19.7 | 13.8 | 6.4 | 2.0 | 0.6 |
| $\mathcal{B}_3(\cdot)$ | 58.6 | 52.9 | 46.4 | 37.7 | 28.8 | 13.9 | 5.0 | 1.3 |
| $\mathcal{B}_4(\cdot)$ | 77.1 | 72.6 | 66.6 | 57.0 | 45.8 | 25.8 | 9.6 | 2.9 |
| $\mathcal{B}_{\text{deg}}(\cdot)$ | 76.6 | 58.9 | 55.6 | 48.9 | 38.5 | 22.1 | 8.7 | 2.8 |
| $\mathcal{B}_{\text{deg+2}}(\cdot)$ | 100.0 | 100.0 | 99.2 | 92.2 | 79.9 | 50.8 | 24.0 | 8.2 |

### F.5 Over-robustness of graph neural networks

#### F.5.1 $\ell_2$-weak

Table 10: Over-robustness $R^{over}$ measured using $\ell_2$-weak (see also Figure 3) with a budget of the degree of the target node. Standard deviations never exceed $1\%$.

| K | 0.1 | 0.5 | 1.0 | 1.5 | 2.0 | 3.0 | 4.0 | 5.0 |
|---|---|---|---|---|---|---|---|---|
| MLP | 43.1% | 35.6% | 30.7% | 25.5% | 19.3% | 9.9% | 3.8% | 1.3% |
| GCN | 35.3% | 30.3% | 25.7% | 19.8% | 14.1% | 5.2% | 1.6% | 0.4% |
| SGC | 37.4% | 29.6% | 23.9% | 20.1% | 14.1% | 5.2% | 2.3% | 0.6% |
| APPNP | 32.5% | 30.1% | 24.3% | 19.6% | 16.8% | 7.0% | 2.4% | 0.9% |
| GAT | 33.3% | 29.2% | 26.6% | 21.2% | 15.7% | 7.3% | 2.9% | 0.9% |
| GATv2 | 33.0% | 29.4% | 26.6% | 21.2% | 16.1% | 7.5% | 2.9% | 0.9% |
| GraphSAGE | 36.7% | 32.3% | 26.8% | 22.2% | 16.9% | 8.3% | 3.3% | 1.2% |
| LP | 16.0% | 13.7% | 11.3% | 8.9% | 6.7% | 3.0% | 1.0% | 0.3% |

### F.5.2 Nettack

Figure 12 and Table 11 shows that over-robustness is not only occurring for weak attacks. Especially, the MLP results show that if we would have a classifier perfectly robust against Nettack in the bounded perturbation set, for all $K \leq 3$ (where structure matters), this would result in high over-robustness.

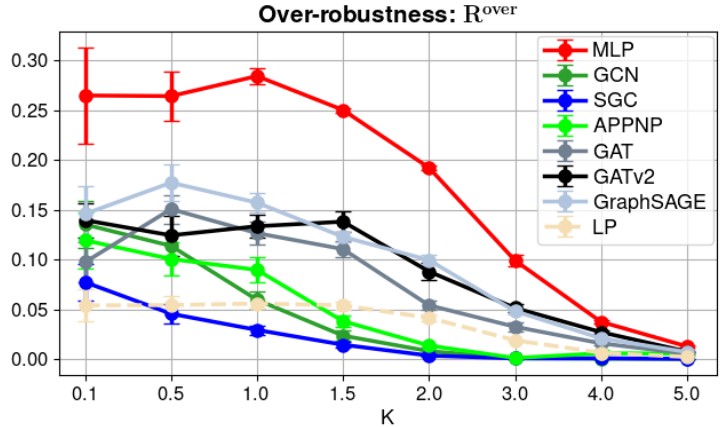

Figure 12: Fraction of robustness beyond semantic change (Nettack, local budget $\Delta$: degree of node). A large part of the measured robustness against $\mathcal{B}_{\deg}(\cdot)$ can be attributed to over-robustness.

Table 11: Over-robustness $R^{over}$ measured using Nettack with a budget of the degree of the target node. Standard deviations rarely exceed $1\%$ notably for MLP at $K = 0.1$ with $4.8\%$.

| K | 0.1 | 0.5 | 1.0 | 1.5 | 2.0 | 3.0 | 4.0 | 5.0 |
|---|---|---|---|---|---|---|---|---|
| MLP | 26.5% | 26.4% | 28.4% | 25.0% | 19.2% | 9.9% | 3.8% | 1.3% |
| GCN | 13.5% | 11.4% | 6.0% | 2.4% | 0.8% | 0.1% | 0.0% | 0.0% |
| SGC | 7.7% | 4.6% | 2.9% | 1.4% | 0.4% | 0.1% | 0.1% | 0.0% |
| APPNP | 11.9% | 10.1% | 9.0% | 3.8% | 1.4% | 0.1% | 0.6% | 0.6% |
| GAT | 9.7% | 15.1% | 12.7% | 11.0% | 5.4% | 3.3% | 1.6% | 0.5% |
| GATv2 | 14.0% | 12.4% | 13.3% | 13.8% | 8.8% | 5.1% | 2.7% | 0.8% |
| GraphSAGE | 14.6% | 17.7% | 15.7% | 12.3% | 9.9% | 4.8% | 2.1% | 0.7% |
| LP | 5.4% | 5.4% | 5.6% | 5.4% | 4.2% | 1.9% | 0.6% | 0.2% |

### F.5.3 DICE

Figure 13 and Table 12 shows, as for Nettack, that over-robustness is not only occurring for $\ell_2$-weak. DICE shows very similar behaviour to $\ell_2$-weak. Especially, the MLP results show that if we would have a classifier perfectly robust against DICE in the bounded perturbation set, for all $K \leq 3$ (where structure matters), this would result in high over-robustness.

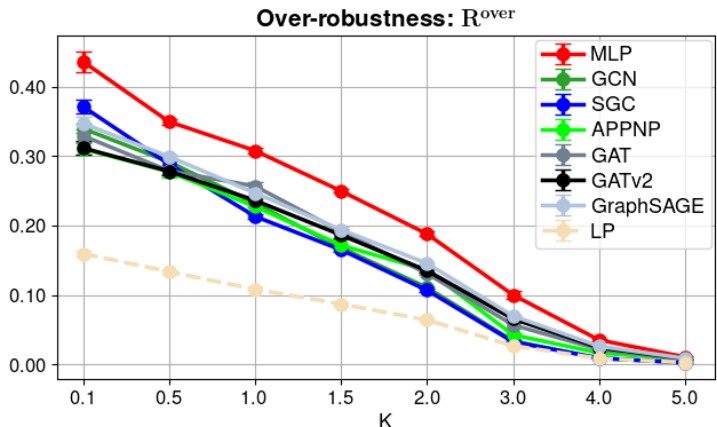

Figure 13: Fraction of robustness beyond semantic change (DICE, local budget $\Delta$: degree of node). A large part of the measured robustness against $\mathcal{B}_{\text{deg}}(\cdot)$ can be attributed to over-robustness.

Table 12: Over-Robustness $R^{over}$ measured using Nettack with a budget of the degree of the target node. Standard deviations are insignificant and removed for brevity.

| K | 0.1 | 0.5 | 1.0 | 1.5 | 2.0 | 3.0 | 4.0 | 5.0 |
|---|-----|-----|-----|-----|-----|-----|-----|-----|
| MLP | 43.6% | 35.0% | 30.8% | 25.0% | 18.8% | 10.0% | 3.5% | 1.1% |
| GCN | 34.0% | 29.2% | 23.2% | 16.8% | 11.1% | 3.4% | 0.9% | 0.3% |
| SGC | 37.2% | 28.9% | 21.3% | 16.5% | 10.7% | 3.3% | 0.9% | 0.2% |
| APPNP | 31.1% | 27.8% | 22.7% | 17.2% | 13.6% | 4.3% | 1.7% | 0.8% |
| GAT | 32.9% | 27.9% | 25.7% | 19.2% | 13.1% | 5.7% | 2.2% | 0.7% |
| GATv2 | 31.2% | 27.8% | 23.6% | 18.7% | 13.5% | 6.6% | 2.4% | 0.8% |
| GraphSAGE | 34.7% | 29.9% | 24.7% | 19.4% | 14.6% | 7.0% | 2.6% | 0.9% |
| LP | 16.0% | 13.4% | 10.8% | 8.7% | 6.4% | 2.7% | 0.9% | 0.3% |

### F.5.4 $\ell_2$-weak: no budget constraints

Figure 14 shows that especially GAT, GATv2 and GraphSAGE and to a lesser extent APPNP have vast areas of over-robustness in input space. This investigation is conceptually motivated by the cases outline in Figure 5 in Appendix A. Note that for high $K$, GAT, GATv2 and GraphSAGE additionally show high adversarial robustness (see Appendix F.6).

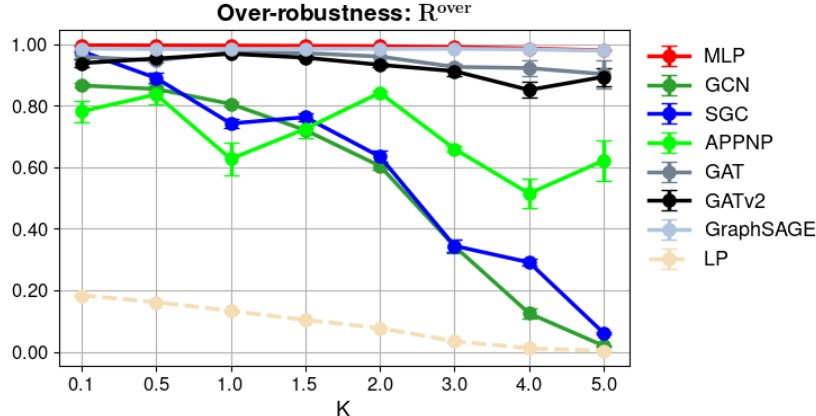

Figure 14: Applying $\ell_2$-weak without budget restriction until it changes a classifiers prediction reveals large over-robust regions in input space (compare conceptually with Figure 5 in Appendix A). Note that label propagation has significantly smaller over-robustness while for $K \leq 1.5$ not achieving worse test accuracy (Appendix F.2) or adversarial robustness (Appendix F.6.1). Especially for high $K$, some architectures additionally show high adversarial robustness (see Appendix F.6).

## F.6 Adversarial robustness of graph neural networks

### F.6.1 Nettack

Nettack is the strongest attack we employ, hence, it is the best heuristic we have to measure (semantic-aware) adversarial robustness $R^{adv}$ (see Metric-Section C) and conventional, non-semantics aware, adversarial robustness $R(f)$ (see Section 4). Figure 15 shows that for $K \leq 1$, LP has not only lowest over-robustness but also best (semantic-aware) adversarial robustness. Some models, for $K \geq 3$ achieve high adversarial robustness, even though they are also highly over-robust (compare with Figure 3 and Appendix F.5.4).

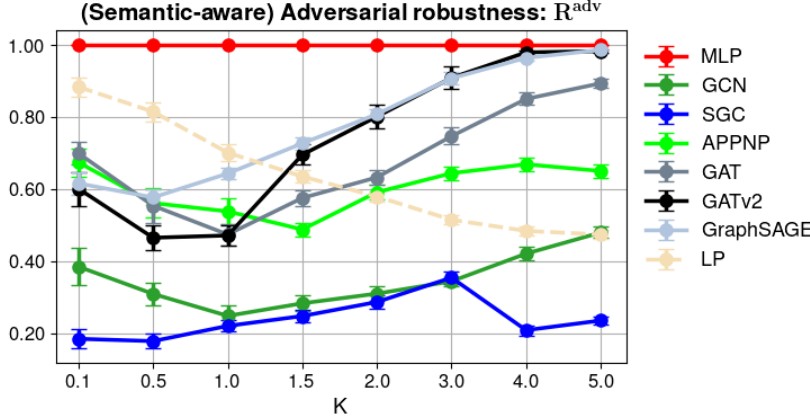

Figure 15: Semantic aware robustness $R^{adv}$ measured using Nettack. Dashed lines are LP models.

Figure 16 shows that similar, albeit slightly less accurate adversarial robustness measurements are obtained by not including the the semantic awareness. Rankings can indeed change, but only if these models are already very close regarding $R^{adv}$ (e.g. LP in truth is more robust than GraphSAGE (Figure 15), but Figure 16 indicates otherwise). This is evidence that if we use a strong attack on a real-world graph, robustness rankings probably stay approximately correct and is good news for using conventional adversarial robustness measures in practice. However, if two methods have very similar robustness scores, conventional adversarial robustness is not accurate enough to differentiate them.

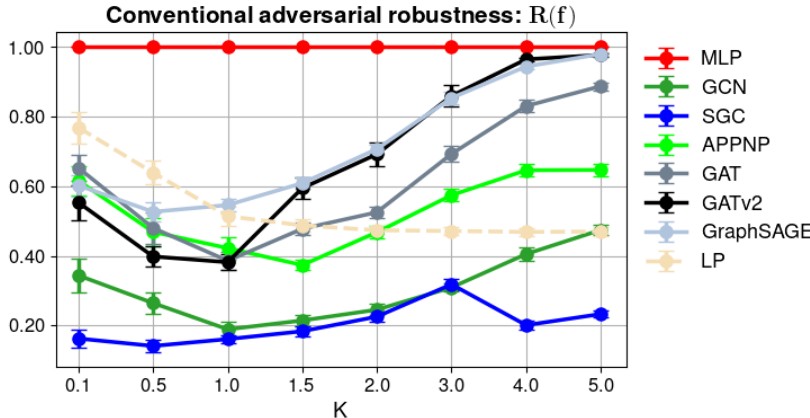

Figure 16: Conventional (degree-corrected) robustness $R^{adv}$ measured using Nettack. Dashed lines are LP models.

The harmonic mean of $R^{adv}$ and $1 - R^{over}$ (see Metric-Section C) shows a complete picture of the robustness of the analysed models. To measure $R^{over}$, we use $\ell_2$-weak. Note that no GNN achieves top placements using this ranking, but the best models, depending on the amount of feature information, are LP or MLP. However, MLP does not have competitive test accuracy expect when structure doesn't matter ($K \geq 4$) (see Appendix F.2).

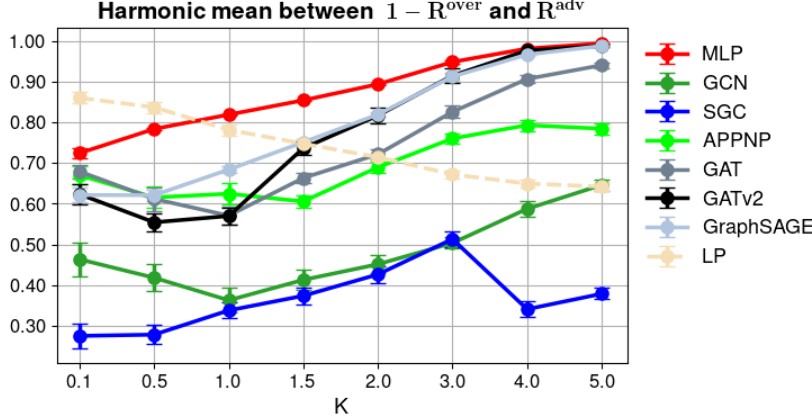

Figure 17: The harmonic mean of $R^{adv}$ and $1 - R^{over}$, with $R^{adv}$ measured using Nettack and $R^{over}$ using $\ell_2$-weak.

### F.6.2 DICE

DICE, as just randomly connecting to different class nodes, turns out to be a very weak attack similar to $\ell_2$-weak. Hence, its adversarial robustness counts differ significantly from the stronger Nettack. Here, we measure a significant difference between true (semantic-aware) adversarial robustness as presented in Figure 18 and conventional adversarial robustness as shown in Figure 19. Indeed, a conventional robustness measurement claims that LP is always the least robust method by significant margins, however, correcting for semantic change uncovers that LP actually is the most robust method for $K \leq 0.5$ and has competitive robustness for $K = 1$ (even until $K = 1.5$ looking at overall robustness 20). We find that with DICE we measure significantly higher over-robustness (Section F.5). Therefore, having a weak attack can result in a significantly different picture of the true compared to conventional adversarial robustness. This gives us insights for applying attacks on real-world graphs, calling for the importance of always choosing a strong, best adaptive attack (Tramèr et al., 2020;

Mujkanovic et al., 2022), if we want to gain insights into the true robustness of a defense and not be "fooled" by over-robust behaviour.

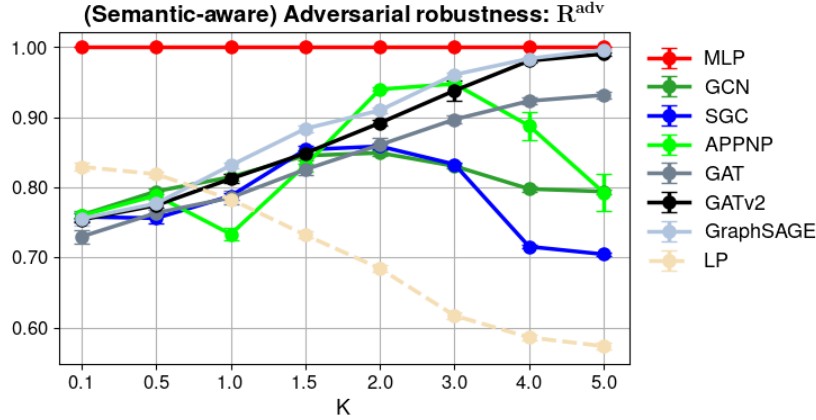

Figure 18: Semantic Aware Robustness $R^{adv}$ measured using DICE. Dashed lines are LP models.

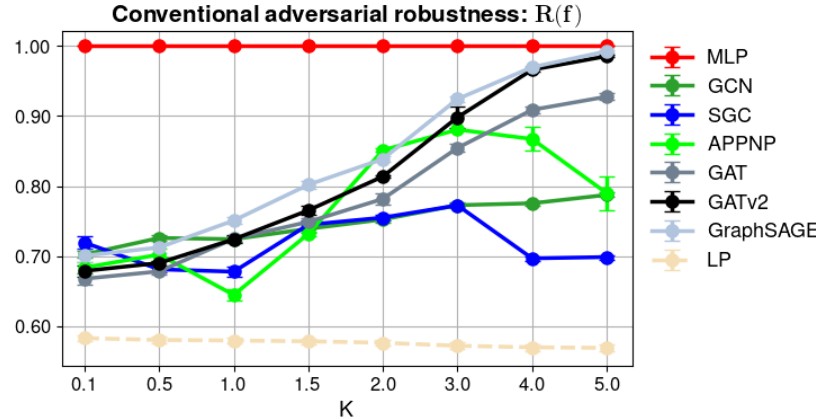

Figure 19: Conventional (Degree-Corrected) Robustness $R^{adv}$ measured using DICE. Dashed lines are LP models.

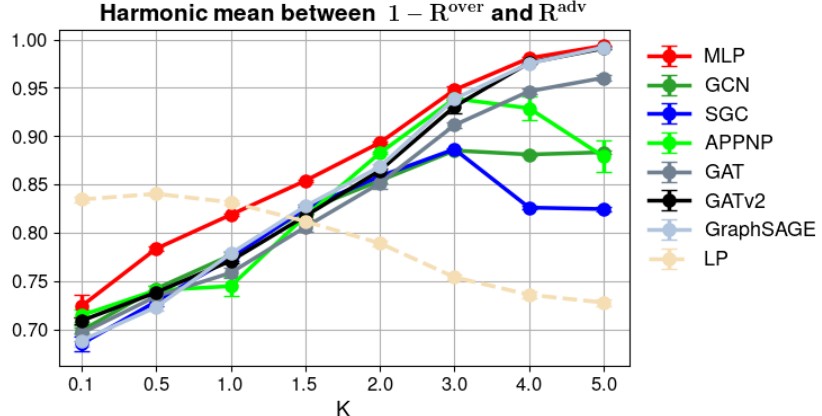

Figure 20: The harmonic mean of $R^{adv}$ and $1 - R^{over}$, with $R^{adv}$ measured using DICE and $R^{over}$ using $\ell_2$-weak.

### F.6.3 Further results on real-world graphs

Table 13: Model test accuracy and standard deviation over eight data splits

| Model dataset | GCN | LP |
|---|---|---|
| Citeseer | $69.4 \pm 1.75$ | $66.5 \pm 2.12$ |
| Cora-ML | $87.1 \pm 2.12$ | $84.0 \pm 2.45$ |

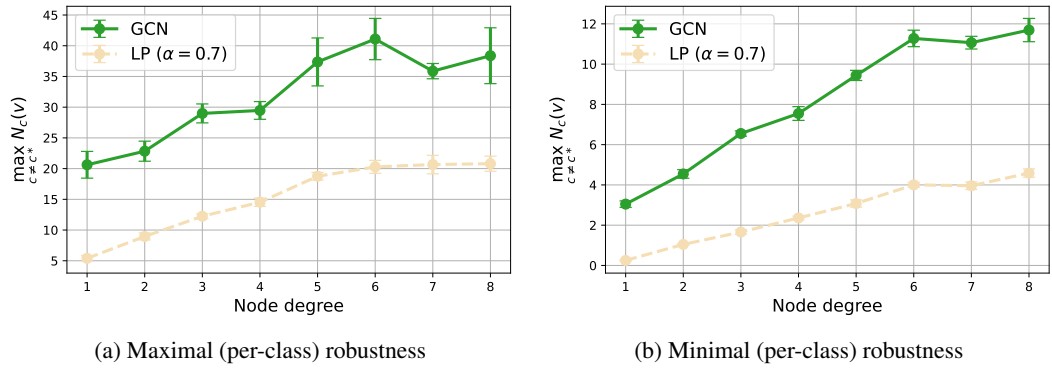

(a) Maximal (per-class) robustness

(b) Minimal (per-class) robustness

Figure 21: Mean robustness per node degree on the Cora-ML dataset. Error bars indicate the standard error of the mean over eight data splits.

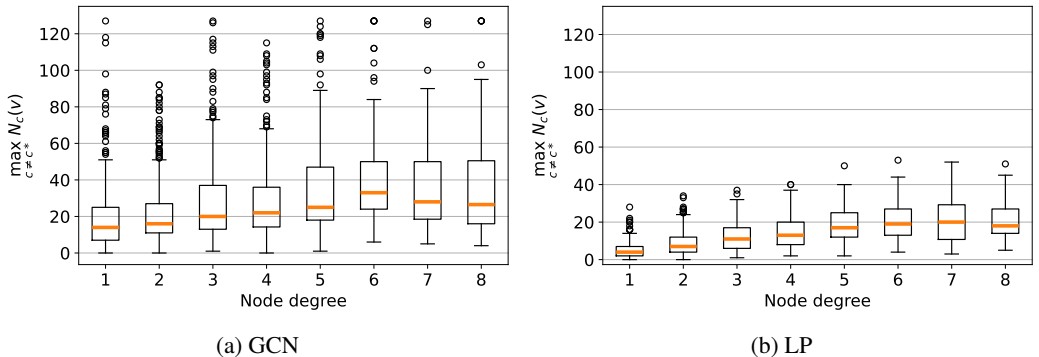

(a) GCN

(b) LP

Figure 22: Distribution of maximal (per-class) node robustness by node degree on the Cora-ML dataset for different models. Results are aggregated over eight data splits.

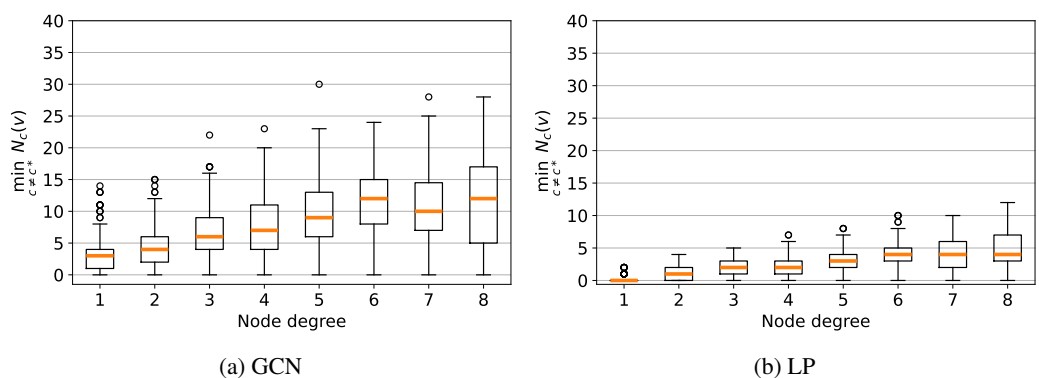

(a) GCN

(b) LP

Figure 23: Distribution of minimal (per-class) node robustness by node degree on the Cora-ML dataset for different models. Results are aggregated over eight data splits.

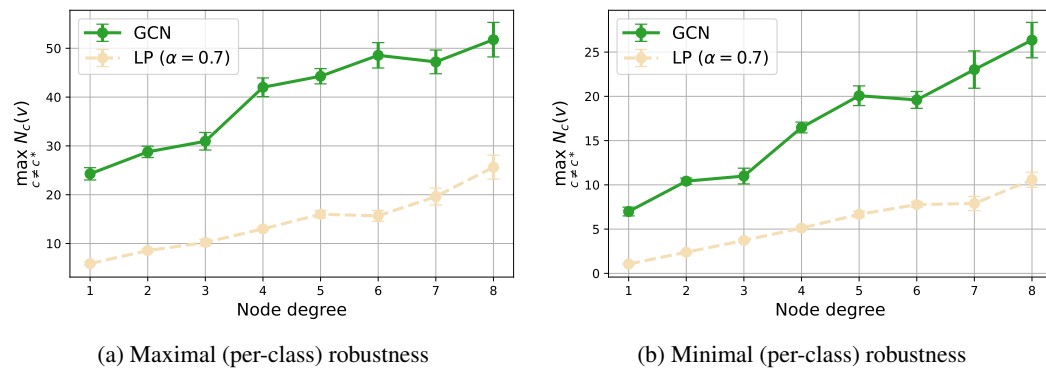

(a) Maximal (per-class) robustness

(b) Minimal (per-class) robustness

Figure 24: Mean robustness per node degree on the Citeseer dataset. Error bars indicate the standard error of the mean over eight data splits.

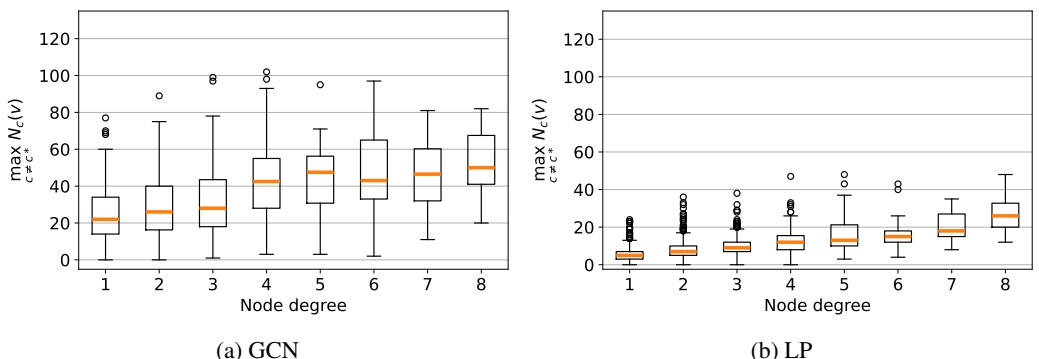

(a) GCN

(b) LP

Figure 25: Distribution of maximal (per-class) node robustness by node degree on the Citeseer dataset for different models. Results are aggregated over eight data splits.

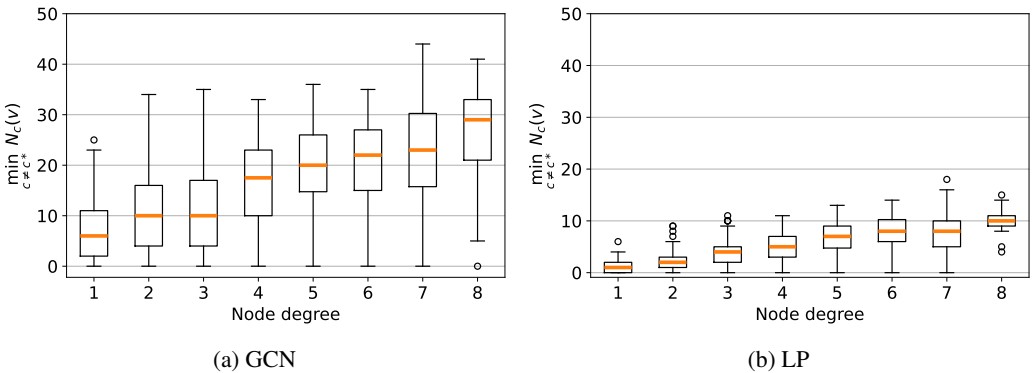

(a) GCN

(b) LP

Figure 26: Distribution of minimal (per-class) node robustness by node degree on the Citeseer dataset for different models. Results are aggregated over eight data splits.

# G  Additional related work

The problem of semantic content preservation was theoretically studied by Suggala et al. (2019). However, they only discuss i.i.d. data and focus on the image domain. To the best of our knowledge, we are the first to study an (adversarial) notion of over-robustness.

We see us related to a new line of work using synthetic graph models to generate principled insights into GNNs. Notably, Fountoulakis et al. (2022) show that in non-trivially CSBMs settings (hard regime), GATs, with high probability, can't distinguish same-class edges from different-class edges and degenerate to GCNs. Baranwal et al. (2021) study GCNs on CSBMs and find graph convolutions extent the linear separability of the data. Palowitch et al. (2022) generate millions of synthetic graphs to explore the performance of common GNNs on graph datasets with different characteristics to the common benchmark real-world graphs. However, their studied degree-corrected SBM is fundamentally limited to transductive learning.

In the graph domain, Dai et al. (2018) note the possibility of measuring semantic content with a gold standard classifier and generate semantic-preserving perturbations for graph-classification on Erdős-Rényi graphs. Regarding sound perturbations models, distantly related is also the work of Geisler et al. (2022), which apply GNNs to combinatorial optimization tasks and therefore, can describe how the perturbations change or preserve the label and thereby, semantics. Although no work has explicitly addressed semantic content preservation for node classification, we list works going beyond $\ell_2$-norm restrictions: Zügner et al. (2018) proposed to approximate unnoticeability by preserving a (power-law) degree distribution; a similar idea to approximatively preserve node-degree distribution is also mentioned in Wang et al. (2018); Li et al. (2021) introduced a metric for degree assortativity, but did not restrict perturbations; Chen et al. (2022) propose other homophily metrics for unnoticeability, but focus on different perturbations to the graph, namely adding malicious nodes.

Regarding the bigger picture in robust graph learning, all works measuring small changes to the graph's structure using the $\ell_0$-norm can be seen as related. This is a large body of work and includes but is not limited to i) the attack literature such as (Zügner et al., 2018; Dai et al., 2018; Waniek et al., 2018; Chen et al., 2018; Zügner & Günnemann, 2019a; Jin et al., 2020; Geisler et al., 2021); ii) various defenses ranging from detecting attacks (Wu et al., 2019b; Entezari et al., 2020), proposing new robust layers and architectures (Zhu et al., 2019; Geisler et al., 2020) to robust training schemes (Zügner & Günnemann, 2019b; Xu et al., 2019, 2020); iii) robust certification (Bojchevski et al., 2020; Schuchardt et al., 2021). An overview of the adversarial robustness literature on GNNs is given by Günnemann (2022). Zheng et al. (2021) provide a graph robustness benchmark.

# H  Discussion

We have shown that on CSBMs, a significant part of conventional robustness of GNNs can be attributed to over-robustness, with similar patterns on real-world graphs. This raises the question, what kind of robustness do defenses improve on in GNNs? As a *positive take-away* for robustness measurements on real-world graphs, we have found that strong attacks lead to conventional adversarial robustness measurements, which are a *good proxy* for semantic-aware adversarial robustness. However, we also found that if the attack is weak, measured conventional robustness can turn out to be over-robustness and hence, semantic-awareness can *significantly* change robustness rankings. We think this calls to practitioners to always choose a strong, best adaptive attack to benchmark robustness of their models and defenses.

Our concept of over-robustness allows to think of a new kind of attack, where the adversary overtakes a clean node (e.g. social media user) and, with its malicious activity, tries to stay undetected. While we have shown that GNNs are highly susceptible to these kind of threats, label propagation is not. As visually inspecting graphs is difficult, we have shown that synthetic graph generation models can be used as important tools to further a principled understanding of graph attacks and defenses. We believe our work outlines a framework for others to build upon.

