# OpenReview forum: "Revisiting Robustness in Graph Machine Learning"
_NeurIPS.cc/2022/Workshop/TSRML — TSRML2022_

### Official Review · Reviewer_dczc · 2022-10-20
**A nice paper**

**Overall Recommendation:** Overall it is a good paper.
**Overall Rating:** 7

**Summary:**

The authors discussed mainstream work on node-level predictions for Graph Neural Networks in this paper. The authors used contextual stochastic block models to enhance the robustness of GNNs. The authors also discussed the risk factors of the robustness of GNN. The authors conduct theoretical work and ablation studies for the robustness of GNNs.

**Strengths:**

1 the authors did good theoretical studies of the robustness of GNN.
2 the authors did excellent experiments and ablation studies for different GNN structures and with different parameters.
3 this paper is well-written,

**Weaknesses:**

The conclusion is trivial.
Minor comments:
The authors should consider citing the following papers
https://arxiv.org/pdf/1810.10751.pdf
In this paper, the authors discussed the small degree nodes are more vulnerable
https://arxiv.org/pdf/2005.10203.pdf
This paper discussed the robustness of GNN for various attacks on structures.


**Review Confidence:**

5: The reviewer is absolutely certain that the evaluation is correct and very familiar with the relevant literature

---

### Official Review · Reviewer_RXVv · 2022-10-21
**Over-robustness in GNNs**

**Overall Rating:** 9

**Summary:**

The paper provides a method to characterize semantic-preserving perturbations on graphs using a reference classifier that provides a way to measure semantic similarity. The authors propose using the data-generating process to build such a reference classifier and demonstrate this for Contextual Stochastic Block Models (CSBMs). The proposed idea not only helps give an intuitive definition of adversarial examples for Graph node prediction problems but also helps define the new and interesting problem of over-robustness of GNNs. The authors define and explore the over-robustness issue for CSBMs but also use the insights derived from the toy models to point out a similar problem in real-world datasets.

**Strengths:**

- The paper provides a novel characterization of semantic similarity among perturbed graphs.
- The authors point out a new and vital problem in graph learning that might be of independent interest to the community.
- The authors propose a new attack method based on the reference classifier that helps understand the robustness problem. The authors suggest some well-motivated new metrics that can be of independent interest.
- The authors use the insights from the artificial data to find indicators for over-robustness that can be generalized to real-world data.
- The paper is generally very well-written and easy to understand.

**Weaknesses:**

- The proposed definition of semantic similarity uses a reference classifier which might be possible to construct for real-world datasets.

**Overall Recommendation:**

The paper contains several good results that would be of great interest to the community. Some of the proposed ideas could help other researchers to form better metrics to measure the robustness of graph neural networks. Some of the insights in the paper could also help develop better GNNs.

**Review Confidence:**

4: The reviewer is confident but not absolutely certain that the evaluation is correct

---

### Decision · Program_Chairs · 2022-10-23

**Decision:**

Accept

**Comment:**

Following the unanimous recommendations from reviewers, the submission is accepted.